# A First-Order Multi-Gradient Algorithm for Multi-Objective Bi-Level Optimization

## Abstract

In this paper, we study the Multi-Objective Bi-Level Optimization (MOBLO) problem, where the upper-level subproblem is a multi-objective optimization problem and the lower-level subproblem is for scalar optimization. Existing gradient-based MOBLO algorithms need to compute the Hessian matrix, causing the computational inefficient problem. To address this, we propose an efficient first-order multi-gradient method for MOBLO, called FORUM. Specifically, we reformulate MOBLO problems as a constrained multi-objective optimization (MOO) problem via the value-function approach. Then we propose a novel multi-gradient aggregation method to solve the challenging constrained MOO problem. Theoretically, we provide the complexity analysis to show the efficiency of the proposed method and a non-asymptotic convergence result. Empirically, extensive experiments demonstrate the effectiveness and efficiency of the proposed FORUM method in different learning problems. In particular, it achieves state-of-the-art performance on three multi-task learning benchmark datasets.

## 1 Introduction

In this work, we study the Multi-Objective Bi-Level Optimization (MOBLO) problem, which is formulated as

$$\min_{\alpha \in \mathbb{R}^n, \omega \in \mathbb{R}^p} F(\alpha, \omega) \quad \text{s.t.} \ \omega \in \mathcal{S}(\alpha) = \arg\min_{\omega} f(\alpha, \omega), \tag{1}$$

where $\alpha$ and $\omega$ denote the Upper-Level (UL) and Lower-Level (LL) variables, respectively. The UL subproblem, $F := (F_1, F_2, \ldots, F_m)^\top : \mathbb{R}^n \times \mathbb{R}^p \to \mathbb{R}^m$, is a vector-valued jointly continuous function for $m$ desired objectives. $\mathcal{S}(\alpha)$ denotes the optimal solution set (which is usually assumed to be a singleton set (Franceschi et al., 2017; Ye et al., 2021)) of the LL subproblem by minimizing a continuous function $f(\alpha, \omega)$ w.r.t. $\omega$. In this work, we focus on MOBLO with a singleton $\mathcal{S}(\alpha)$ and a non-convex UL subproblem, where $F_i$ is a non-convex function for all $i$. MOBLO has demonstrated its superiority in various learning problems such as neural architecture search (Elsken et al., 2018; Lu et al., 2020; Liu & Jin, 2021; Yue et al., 2022), reinforcement learning (Chen et al., 2019; Yang et al., 2019; Abdolmaleki et al., 2020), multi-task learning (Ye et al., 2021; Mao et al., 2022), and meta-learning (Ye et al., 2021; Yu et al., 2023).

Recently, MOML (Ye et al., 2021) and MoCo (Fernando et al., 2023) are proposed as effective gradient-based MOBLO algorithms, which hierarchically optimize the UL and LL variables based on ITerative Differentiation (ITD) based Bi-Level Optimization (BLO) approach (Maclaurin et al., 2015; Franceschi et al., 2017; 2018; Grazzi et al., 2020). Specifically, given $\alpha$, both MOML and MoCo first compute the LL solution $\omega^*(\alpha)$ by solving LL subproblem with $T$ iterations and then update $\alpha$ via the combination of the hypergradients $\{\nabla_\alpha F_i(\alpha, \omega^*(\alpha))\}_{i=1}^m$. Note that they need to calculate the complex gradient $\nabla_\alpha \omega^*(\alpha)$, which requires to compute many Hessian-vector products via the chain rule. Besides, their time and memory costs grow significantly fast with respect to the dimension of $\omega$ and $T$. Therefore, existing gradient-based methods to solve MOBLO problems could suffer from the inefficiency problem, especially in deep neural networks.

To address this limitation, we propose an efficient **F**irst-**O**rde**R** m**U**lti-gradient method for **M**OBLO (**FORUM**). Specifically, we reformulate MOBLO as an equivalent constrained multi-objective optimization (MOO) problem by the value-function-based approach (Liu et al., 2021c; 2022a; Sow et al., 2022). Then, we propose a multi-gradient aggregation method to solve the challenging

constrained MOO problem. Different from MOML and MoCo, FORUM is a fully first-order algorithm and does not need to calculate the high-order Hessian matrix. Theoretically, we provide the complexity analysis showing that FORUM is more efficient than MOML and MoCo in both time and memory costs. In addition, we provide a non-asymptotic convergence analysis for FORUM. Empirically, we evaluate the effectiveness and efficiency of FORUM on two learning problems, i.e., multi-objective data hyper-cleaning and multi-task learning on three benchmark datasets.

The main contributions of this work are three-fold:

- We propose the FORUM method, an efficient gradient-based algorithm for the MOBLO problem;
- We demonstrate FORUM is more efficient than existing MOBLO methods from the perspective of complexity analysis and provide a non-asymptotic convergence analysis;
- Extensive experiments demonstrate the effectiveness and efficiency of the proposed FORUM method. In particular, it achieves state-of-the-art performance on three benchmark datasets under the setting of multi-task learning.

## 2 RELATED WORKS

**Multi-Objective Optimization.** MOO aims to solve multiple objectives simultaneously and its goal is to find the Pareto-optimal solution. MOO algorithms can be divided into three categories: population-based (Angus, 2007), evolutionary-based (Zhou et al., 2011), and gradient-based (Désidéri, 2012; Mahapatra & Rajan, 2020). In this paper, we focus on the last category. MGDA algorithm (Désidéri, 2012) is a representative gradient-based MOO method, which finds a gradient update direction to make all the objectives decrease in every training iteration by solving a quadratic programming problem. Compared with the widely-used linear scalarization approach which linearly combines multiple objectives to a single objective, MGDA and its variants (Fernando et al., 2023; Zhou et al., 2022) have shown their superiority in many learning problems such as multi-task learning (Sener & Koltun, 2018) and reinforcement learning (Yu et al., 2020), especially when some objectives are conflicting.

**Bi-Level Optimization.** BLO (Liu et al., 2021b) is a type of optimization problem with a hierarchical structure, where one subproblem is nested within another subproblem. The MOBLO problem (1) reduces degrades to BLO problem when $m$ equals 1. One representative category of the BLO method is the ITD-based methods (Maclaurin et al., 2015; Franceschi et al., 2017; 2018; Grazzi et al., 2020) that use approximated hypergradient to optimize the UL variable, which is computed by the automatic differentiation based on the optimization trajectory of the LL variable. Some value-function-based algorithms (Liu et al., 2022a; 2021c; Sow et al., 2022) have been proposed recently to solve BLO by reformulating the original BLO to an equivalent optimization problem with a simpler structure. The value-function-based reformulation strategy naturally yields a first-order algorithm, hence it has high computational efficiency.

**Multi-Objective Bi-Level Optimization.** MOML (Ye et al., 2021) is proposed as the first gradient-based MOBLO algorithm. However, MOML needs to calculate the complex Hessian matrix to obtain the hypergradient, causing the computationally inefficient problem. MoCo (Fernando et al., 2023) also employs the ITD-based approach like MOML for hypergradient calculation. It uses a momentum-like gradient approximation approach for hypergradient and a one-step approximation method to update the weights. It has the same inefficiency problem as the MOML method. Yu et al. (2023) propose a mini-batch approach to optimize the UL subproblem in the MOBLO. However, it aims to generate weights for a huge number of UL objectives and is different from what we focus on. MORBiT (Gu et al., 2023) studies a BLO problem with multiple objectives in its UL subproblem but it formulates the UL subproblem as a min-max problem, which is different from problem (1) we focus on in this paper.

## 3 THE FORUM ALGORITHM

In this section, we introduce the proposed FORUM method. Firstly, we reformulate MOBLO as an equivalent constrained multi-objective problem via the value-function-based approach in Section 3.1.

Next, we provide a novel multi-gradient aggregation method to solve the constrained multi-objective problem in Section 3.2.

## 3.1 REFORMULATION OF MOBLO

Based on the value-function-based approach (Liu et al., 2021c; 2022a; Sow et al., 2022; Kwon et al., 2023), we reformulate MOBLO problem (1) as an equivalent single-level *constrained multi-objective optimization* problem:

$$\min_{\alpha \in \mathbb{R}^n, \omega \in \mathbb{R}^p} F(\alpha, \omega) \quad \text{s.t.} \ f(\alpha, \omega) \le f^*(\alpha), \tag{2}$$

where $f^*(\alpha) = \min_\omega f(\alpha, \omega) = f(\alpha, \omega^*(\alpha))$ is the *value function*, which represents the lower bound of $f(\alpha, \omega)$ w.r.t. $\omega$. To simplify the notation, we define $z \equiv (\alpha, \omega) \in \mathbb{R}^{n+p}$ and $\mathcal{Z} \equiv \mathbb{R}^n \times \mathbb{R}^p$. Then, we have $F(z) \equiv F(\alpha, \omega)$ and $f(z) \equiv f(\alpha, \omega)$. Thus, problem (2) can be rewritten as

$$\min_{z \in \mathcal{Z}} F(z) \quad \text{s.t.} \ q(z) \le 0, \tag{3}$$

where $q(z) = f(z) - f^*(\alpha)$ is the *constraint function*. Since the gradient of the value function $f^*(\alpha)$ is $\nabla_\alpha f^*(\alpha) = \nabla_\alpha f(\alpha, \omega^*(\alpha)) = \nabla_\alpha f(\alpha, \omega^*)$ by the chain rule and $\nabla_\omega f(\alpha, \omega)\mid_{\omega = \omega^*(\alpha)} = 0$, we do not need to compute the complex Hessian matrix $\nabla_\alpha \omega^*(\alpha)$ like MOML and MoCo.

However, solving problem (3) is challenging for two reasons. One reason is that the Slater's condition (Chen et al., 2023), which is required for duality-based optimization methods, does not hold for problem (3), since the constraint $q(z) \le 0$ is ill-posed (Liu et al., 2021c; Jiang et al., 2023) and does not have an interior point. To see this, we assume $z_0 = (\alpha_0, \omega_0) \in \mathcal{Z}$ and $q(z_0) \le 0$. Then the constraint $q(z) \le 0$ is hard to be satisfied at the neighborhood of $\alpha_0$, unless $f^*(\alpha)$ is a constant function around $\alpha_0$, which rarely happens. Therefore, problem (3) cannot be treated as classic constrained optimization and we propose a novel gradient method to solve it in Section 3.2. Another reason is that for given $\alpha$, the computation of $\omega^*(\alpha)$ is intractable. Thus, we approximate it by $\tilde{\omega}^T$ computed by $T$ steps of gradient descent. Specifically, given $\alpha$ and an initialization $\tilde{\omega}^0$ of $\omega$, we have

$$\tilde{\omega}^{t+1} = \tilde{\omega}^t - \eta \nabla_\omega f(\alpha, \tilde{\omega}^t), \quad t = 0, \cdots, T-1, \tag{4}$$

where $\eta$ is the step size. Then, the constraint function $q(z)$ is approximated by $\tilde{q}(z) = f(z) - f(\alpha, \tilde{\omega}^T)$ and its gradient $\nabla_z q(z)$ is approximated by $\nabla_z \tilde{q}(z)$. We show that the approximation error of gradient exponentially decays w.r.t. the LL iterations $T$ in Appendix A.1. Hence, problem (3) is modified to

$$\min_{z \in \mathcal{Z}} F(z) \quad \text{s.t.} \ \tilde{q}(z) = f(z) - f(\alpha, \tilde{\omega}^T) \le 0. \tag{5}$$

## 3.2 MULTI-GRADIENT AGGREGATION METHOD

In this section, we introduce the proposed multi-gradient aggregation method for solving problem (5) iteratively. Specifically, at $k$-th iteration, assume $z_k$ is updated by $z_{k+1} = z_k + \mu d_k$ where $\mu$ is the step size and $d_k$ is the update direction for $z_k$. Then, we expect $d_k$ can simultaneously minimize the UL objective $F(z)$ and the constraint function $\tilde{q}(z)$. Note that the minimum of the approximated constraint function $\tilde{q}(z)$ converges to the minimum of $q(z)$, i.e. 0, as $T \to +\infty$. Thus, we expect $d_k$ to decrease $\tilde{q}(z)$ consistently such that the constraint $\tilde{q}(z) \le 0$ is satisfied.

Note that there are multiple potentially conflicting objectives $\{F_i\}_{i=1}^m$ in the UL subproblem. Hence, we expect $d_k$ can decrease every objective $F_i$, which can be formulated as the following problem to find $d_k$ to maximize the minimum decrease across all objectives as

$$\max_d \min_{i \in [m]} (F_i(z_k) - F_i(z_k + \mu d)) \approx -\mu \min_d \max_{i \in [m]} \langle \nabla F_i(z_k), d \rangle. \tag{6}$$

To regularize the update direction, we add a regularization term $\frac{1}{2}\|d\|^2$ to problem (6) and compute $d_k$ by solving $\min_d \max_{i \in [m]} \langle \nabla F_i(z_k), d \rangle + \frac{1}{2}\|d\|^2$.

To decrease the constraint function $\tilde{q}(z)$, we expect the inner product of $-d$ and $\nabla \tilde{q}(z_k)$ to hold positive during the optimization process, i.e., $\langle \nabla \tilde{q}(z_k), -d \rangle \ge \phi$, where $\phi$ is non-negative constant. To further guarantee that $\tilde{q}(z)$ can be optimized such that the constraint $\tilde{q}(z) \le 0$ can be satisfied, we introduce a dynamic $\phi_k$ here. Specifically, inspired by Gong et al. (2021), we set $\phi_k = \frac{\rho}{2}\|\nabla \tilde{q}(z_k)\|^2$, where $\rho$ is a positive constant. When $\phi_k > 0$, it means that $\|\nabla \tilde{q}(z)\| \ne 0$ and $\tilde{q}(z)$ should be further

---

**Algorithm 1** The FORUM Method

---

**Require:** number of iterations $(K, T)$, step size $(\mu, \eta)$, coefficient $\beta_k$, constant $\rho$
 1: Randomly initialize $z_0 = (\alpha_0, \omega_0)$;
 2: Initialize $\tilde{\lambda}_i^{-1} = 0$, $i = 1, ..., m$;
 3: **for** $k = 0$ **to** $K - 1$ **do**
 4:  Set $\tilde{\omega}^0 = \omega_0$ or $\tilde{\omega}^0 = \omega_k$;
 5:  **for** $t = 0$ **to** $T - 1$ **do**
 6:   Update $\tilde{\omega}$ as $\tilde{\omega}^{t+1} = \tilde{\omega}^t - \eta \nabla_\omega f(\alpha_k, \tilde{\omega}^t)$;
 7:  **end for**
 8:  Set $\tilde{q}(z_k) = f(z_k) - f(\alpha_k, \tilde{\omega}^T)$;
 9:  Compute gradient $\nabla_z \tilde{q}(z_k) = \nabla_z f(z_k) - \nabla_\alpha f(\alpha_k, \tilde{\omega}^T)$;
10:  Compute gradients $\nabla_z F_i(z_k)$, $i = 1, \ldots, m$;
11:  Compute $\lambda^k$ by solving problem (11);
12:  Compute the momentum update $\tilde{\lambda}^k = (1 - \beta_k)\tilde{\lambda}^{k-1} + \beta_k \lambda^k$;
13:  Compute $\nu(\tilde{\lambda}^k)$ via Eq. (9);
14:  Compute $d_k$ via Eq. (8);
15:  Update $z$ as $z_{k+1} = z_k + \mu d_k$;
16: **end for**
17: **return** $z_K$.

---

optimized, and $\langle \nabla \tilde{q}(z_k), -d \rangle \geq \phi_k > 0$ can enforce $\tilde{q}(z)$ to decrease. When $\phi_k$ equals 0, it indicates that the optimum of $\tilde{q}(z)$ is reached and $\langle \nabla \tilde{q}(z_k), -d \rangle \geq \phi_k = 0$ also holds. Thus, the dynamic $\phi_k$ can ensure $d_k$ to iteratively decrease $\tilde{q}(z)$ such that the constraint $\tilde{q}(z) \leq 0$ is satisfied.

Therefore, at $k$-th iteration, we can find $d_k$ by solving the following problem,

$$\min_d \max_{i \in [m]} \langle \nabla F_i(z_k), d \rangle + \frac{1}{2}\|d\|^2, \quad \text{s.t.} \ \langle \nabla \tilde{q}(z_k), d \rangle \leq -\phi_k. \tag{7}$$

Based on the Lagrangian multiplier method, problem (7) has a solution as

$$d_k = -\left( \sum_{i=1}^m \lambda_i^k \nabla F_i(z_k) + \nu(\lambda^k) \nabla \tilde{q}(z_k) \right), \tag{8}$$

where Lagrangian multipliers $\lambda^k = (\lambda_1^k, \ldots, \lambda_m^k) \in \Delta^{m-1}$ (i.e., $\sum_{i=1}^m \lambda_i^k = 1$ and $\lambda_i^k \geq 0$) and $\nu(\lambda)$ is a function of $\lambda$ as

$$\nu(\lambda) = \max\left( \sum_{i=1}^m \lambda_i \pi_i(z_k), 0 \right) \quad \text{with} \ \pi_i(z_k) = \frac{2\phi_k - \langle \nabla \tilde{q}(z_k), \nabla F_i(z_k) \rangle}{\|\nabla \tilde{q}(z_k)\|^2}. \tag{9}$$

Here $\lambda_i^k$ can be obtained by solving the following dual problem as

$$\lambda^k = \arg\min_{\lambda \in \Delta^{m-1}} \frac{1}{2} \left\| \sum_{i=1}^m \lambda_i \nabla F_i(z_k) + \nu(\lambda) \nabla \tilde{q}(z_k) \right\|^2 - \nu(\lambda)\phi_k. \tag{10}$$

The detailed derivations of the above procedure are put in Appendix A.2. Problem (10) can be reformulated as

$$\min_{\lambda \in \Delta^{m-1}, \gamma} \frac{1}{2} \left\| \sum_{i=1}^m \lambda_i \nabla F_i(z_k) + \gamma \nabla \tilde{q}(z_k) \right\|^2 - \gamma\phi_k \quad \text{s.t.} \ \gamma \geq 0, \gamma \geq \sum_{i=1}^m \lambda_i \pi_i(z_k). \tag{11}$$

The first term of the objective function in problem (11) can be simplified to $R^\top \Lambda^\top \Lambda R$, where $R = (\lambda_1, \ldots, \lambda_m, \gamma)^\top$ and $\Lambda = (\nabla F_1, \ldots, \nabla F_m, \nabla \tilde{q})$. Note that the dimension of the matrix $\Lambda^\top \Lambda$ is $(m+1) \times (m+1)$, which is independent with the dimension of $z$. As the number of UL objectives $m$ is usually small, solving problem (11) does not incur too much computational cost. In practice, we can use the open-source CVXPY library (Diamond & Boyd, 2016) to solve problem (11).

To ensure convergence, the sequence of $\{\lambda^k\}_{k=1}^K$ should be a convergent sequence (refer to the discussion in Appendix A.3). However, $\{\lambda^k\}_{k=1}^K$ obtained by directly solving the problem (11) in each

iteration cannot ensure such properties. Therefore, we apply a momentum strategy (Zhou et al., 2022) to $\lambda$ to generate a stable sequence and further guarantee the convergence. Specifically, in $k$-th iteration, we first solve the problem (11) to obtain $\lambda^k$, then update the weights by $\tilde{\lambda}^k = (1 - \beta_k)\tilde{\lambda}^{k-1} + \beta_k \lambda^k$, where $\beta_k \in (0, 1]$ is set to 1 at the beginning and asymptotically convergent to 0 as $k \to +\infty$.

After obtaining $\tilde{\lambda}^k$ with the momentum update, we can compute the corresponding $\nu(\tilde{\lambda}^k)$ via Eq. (9). Then we obtain the update direction $d_k$ by Eq. (8) and update $z_k$ as $z_{k+1} = z_k + \mu d_k$. The entire FORUM algorithm is shown in Algorithm 1.

## 4 ANALYSIS

In this section, we provide complexity analysis and convergence analysis for the FORUM method.

### 4.1 COMPLEXITY ANALYSIS

For the proposed FORUM method, it takes time $\mathcal{O}(pT)$ and space $\mathcal{O}(p)$ to obtain $\widetilde{q}(z)$, and then the computations of all the gradients including $\nabla_z F_i(z)$ and $\nabla_z \widetilde{q}(z)$ require time $\mathcal{O}((n+p)(m+1))$ and space $\mathcal{O}((n+p)(m+1))$. When $m \ll \min\{n, p\}$, the time and space costs of solving the quadratic programming problem (11), which only depends on $m$, can be negligible. Therefore, FORUM runs in time $\mathcal{O}(mn + p(m + T))$ and space $\mathcal{O}(mn + mp)$ in total for each UL iteration.

For existing MOBLO methods (i.e., MOML and MoCo), after $T$-iteration update for the LL subproblem in time $\mathcal{O}(pT)$ and space $\mathcal{O}(p)$, calculating the Hessian-matrix product via backward propagation can be evaluated in time $\mathcal{O}(p(n + p)T)$ and space $\mathcal{O}(n + pT)$. Similar to FORUM, the cost of solving the quadratic programming problem in MOML is also negligible. Note that MoCo applies a momentum update to the UL variables, which causes an additional $\mathcal{O}(mn)$ space cost. Thus, for each UL iteration, MOML and MoCo require $\mathcal{O}(mp(n + p)T)$ time in total, and they require $\mathcal{O}(mn + mpT)$ and $\mathcal{O}(2mn + mpT)$ space, respectively.

In summary, the above analysis indicates FORUM is more efficient than MOML and MoCo in terms of time and space complexity.

### 4.2 CONVERGENCE ANALYSIS

In this section, we analyze the convergence property of FORUM. Firstly, we make an assumption for the UL subproblem.

**Assumption 4.1.** For $i = 1, \ldots, m$, it is assumed that $\nabla F_i(\alpha, \omega)$ is $L_F$-Lipschitz continuous with respect to $z := (\alpha, \omega)$. The $\ell_2$ norm of $\nabla F_i(z)$ and $|F_i(z)|$ are upper-bounded by a constant $M$.

The smoothness and the boundedness assumptions in Assumption 4.1 are widely adopted in non-convex multi-objective optimization (Zhou et al., 2022; Fernando et al., 2023). Then we make an assumption for the LL subproblem.

**Assumption 4.2.** The function $f(\alpha, \omega)$ is $c$-strongly convex with respect to $\omega$, and $\nabla f(\alpha, \omega)$ is $L_f$-Lipschitz continuous with respect to $z := (\alpha, \omega)$.

The strongly convexity assumption in Assumption 4.2 is commonly used in the analysis for the BLO (Maclaurin et al., 2015; Franceschi et al., 2017; 2018) and MOBLO problems (Fernando et al., 2023; Ye et al., 2021). The proposed FORUM algorithm focuses on generating one Karush-Kuhn-Tucker (KKT) stationary point of the original constrained multi-objective optimization problem (3). Following (Gong et al., 2021; Liu et al., 2022a), we measure the convergence of problem (3) by both its KKT stationary condition and the feasibility condition, where detailed definitions are provided in Appendix B.1. Specifically, we denote by $\mathcal{K}(z_k) = \left\| \sum_{i=1}^m \tilde{\lambda}_i^k \nabla F_i(z_k) + \nu_k \nabla q(z_k) \right\|^2$ the measure of KKT stationary condition in the $k$-th iteration, where $\nu_k = \nu(\tilde{\lambda}^k)$. To satisfy the feasibility condition of problem (3), the non-negative function $q(z_k)$ should decrease to 0. Then, with a non-convex multi-objective UL subproblem, we have the following convergence result.

**Theorem 4.3.** *Suppose that Assumptions 4.1 and 4.2 hold, and the sequence $\{z_k\}_{k=0}^K$ generated by Algorithm 1 satisifes $q(z_k) \leq B$, where $B$ is a positive constant. Then if $\eta \leq 1/L_f$, $\mu = \mathcal{O}(K^{-1/2})$,*

*and $\beta = \mathcal{O}(K^{-3/4})$, there exists a constant $C > 0$ such that when $T \geq C$, for any $K > 0$, we have*

$$\max\left\{\min_{k<K} \mathcal{K}(z_k), q(z_k)\right\} = \mathcal{O}(K^{-1/4} + \Gamma(T)), \tag{12}$$

*where $\Gamma(T)$ represents exponential decays with respect to $T$.*

The proof is put in Appendix B.3. Theorem 4.3 gives a non-asymptotic convergence result for Algorithm 1, which depends on both numbers of steps in the UL and LL subproblems (i.e., $K$ and $T$).

## 5 EXPERIMENTS

In this section, we empirically evaluate the proposed FORUM method on different learning problems. All experiments are conducted on a single NVIDIA GeForce RTX 3090 GPU.

### 5.1 DATA HYPER-CLEANING

**Setup.** Data hyper-cleaning (Bao et al., 2021; Franceschi et al., 2017; Liu et al., 2022a;b; Shaban et al., 2019) is a specific hyperparameter optimization problem, where a model is trained on a dataset with part of corrupted training labels. Thus, it aims to reduce the influence of noisy examples by learning to weigh the train samples in a bi-level optimization manner. Here we extend it to a multi-objective setting, where we aim to train a model on multiple corrupted datasets.

Specifically, suppose that there are $m$ corrupted datasets. $\mathcal{D}_i^{\mathrm{tr}} = \{x_{i,j}, y_{i,j}\}_{j=1}^{N_i}$ and $\mathcal{D}_i^{\mathrm{val}}$ denote the noisy training set and the clean validation set for the $i$-th dataset, respectively, where $x_{i,j}$ denotes the $j$-th training sample in the $i$-th dataset, $y_{i,j}$ is the corresponding label, and $N_i$ denotes the size of the $i$-th training dataset. Let $\omega$ denote the model parameters and $\alpha_{i,j}$ denotes the weight of $x_{i,j}$. Let $\mathcal{L}_i^{\mathrm{val}}(\omega; \mathcal{D}_i^{\mathrm{val}})$ be the average loss of model $\omega$ on the clean validation set of the $i$-th dataset and $\mathcal{L}_i^{\mathrm{tr}}(\alpha, \omega; \mathcal{D}_i^{\mathrm{tr}}) = \frac{1}{N_i}\sum_{j=1}^{N_i} \sigma(\alpha_{i,j})\ell(\omega; x_{i,j}, y_{i,j})$ be the weighted average loss on the noisy training set of the $i$-th dataset, where $\sigma(\cdot)$ is an element-wise sigmoid function to constrain each weight in the range $[0, 1]$ and $\ell(\omega; x, y)$ denotes the loss of model $\omega$ on sample $(x, y)$. Therefore, the objective function of this multi-objective data hyper-cleaning is formulated as

$$\min_{\alpha, \omega} \left(\mathcal{L}_1^{\mathrm{val}}(\omega; \mathcal{D}_1^{\mathrm{val}}), \cdots, \mathcal{L}_m^{\mathrm{val}}(\omega; \mathcal{D}_m^{\mathrm{val}})\right)^\top \quad \text{s.t.} \quad \omega \in \mathcal{S}(\alpha) = \arg\min_\omega \sum_{i=1}^m \mathcal{L}_i^{\mathrm{tr}}(\alpha, \omega; \mathcal{D}_i^{\mathrm{tr}}).$$

**Datasets.** We conduct experiments on the MNIST (LeCun et al., 1998) and FashionMNIST (Xiao et al., 2017) datasets. Each dataset corresponds to a 10-class image classification problem. All the images have the same size of $28 \times 28$. Following Bao et al. (2021), we randomly sample 5000, 1000, 1000, and 5000 images from each dataset as the training set, validation set 1, validation set 2, and test set, respectively. The training set and validation set 1 are used to formulate the LL and UL subproblems, respectively. The validation set 2 is used to select the best model and the testing evaluation is conducted on the test set. Half of the samples in the training set are contaminated by assigning them to another random class. *Due to page limit, implementation details are put in Appendix D.1.*

**Results.** Table 1 shows the results on both two datasets under different numbers of LL iterations (i.e., $T = 16, 32, 64, 128$). The classification accuracy and F1 score computed on the test set are used as the evaluation metrics. As can be seen, the proposed FORUM method outperforms the MOML and MoCo in all the settings, which demonstrates the effectiveness of the proposed FORUM method.

Figures 1(a) and 1(b) show that MOML and MoCo need longer running time than FORUM in every configuration of the UL iteration $T$ and the number of LL parameters $p$, respectively, which implies FORUM has a lower time complexity. Figures 1(c) and 1(d) show the change of memory cost per iteration with respect to the LL iteration $T$ and the number of LL parameters $p$, respectively. As can be seen, the memory cost remains almost constant with different $T$'s for FORUM and increases faster for MOML and MoCo. Moreover, the memory cost slightly increases in FORUM with increasing $p$, while it linearly increases in MOML and MoCo. In summary, the results in Figure 1 match the complexity analysis in Section 4.1 and demonstrate that FORUM is more efficient than MOML and MoCo.

Table 1: Performance of different methods with different numbers of LL iterations $T$ on the MNIST and FashionMNIST datasets for the multi-objective data hyper-cleaning problem. Each experiment is repeated over 3 random seeds, and the mean as well as the standard deviation is reported. The best result for each $T$ is marked in **bold**.

| $T$ | Methods | MNIST | | FashionMNIST | |
| --- | --- | --- | --- | --- | --- |
| | | Accuracy (%) | F1 Score | Accuracy (%) | F1 Score |
| 16 | MOML | $88.81_{\pm0.17}$ | $88.78_{\pm0.16}$ | $79.98_{\pm0.21}$ | $79.59_{\pm0.40}$ |
| | MoCo | $88.25_{\pm0.31}$ | $88.22_{\pm0.30}$ | $80.09_{\pm0.25}$ | $79.65_{\pm0.59}$ |
| | FORUM (**ours**) | $\mathbf{90.79}_{\pm0.33}$ | $\mathbf{90.79}_{\pm0.33}$ | $\mathbf{82.37}_{\pm1.00}$ | $\mathbf{82.10}_{\pm1.16}$ |
| 32 | MOML | $87.29_{\pm0.72}$ | $87.26_{\pm0.71}$ | $80.63_{\pm0.58}$ | $80.50_{\pm0.28}$ |
| | MoCo | $87.59_{\pm0.42}$ | $87.56_{\pm0.42}$ | $80.42_{\pm0.47}$ | $80.41_{\pm0.14}$ |
| | FORUM (**ours**) | $\mathbf{90.65}_{\pm0.44}$ | $\mathbf{90.63}_{\pm0.47}$ | $\mathbf{82.11}_{\pm0.72}$ | $\mathbf{81.79}_{\pm1.01}$ |
| 64 | MOML | $88.64_{\pm0.94}$ | $88.61_{\pm0.98}$ | $80.64_{\pm0.35}$ | $80.60_{\pm0.49}$ |
| | MoCo | $88.05_{\pm1.21}$ | $88.03_{\pm1.27}$ | $80.94_{\pm0.19}$ | $80.67_{\pm0.25}$ |
| | FORUM (**ours**) | $\mathbf{90.81}_{\pm0.14}$ | $\mathbf{90.81}_{\pm0.15}$ | $\mathbf{82.07}_{\pm0.38}$ | $\mathbf{81.72}_{\pm0.57}$ |
| 128 | MOML | $88.88_{\pm0.33}$ | $88.86_{\pm0.36}$ | $80.31_{\pm0.45}$ | $80.10_{\pm0.33}$ |
| | MoCo | $88.21_{\pm0.33}$ | $88.20_{\pm0.36}$ | $80.31_{\pm0.30}$ | $79.81_{\pm0.50}$ |
| | FORUM (**ours**) | $\mathbf{90.13}_{\pm0.37}$ | $\mathbf{90.11}_{\pm0.36}$ | $\mathbf{82.07}_{\pm0.73}$ | $\mathbf{81.79}_{\pm0.97}$ |

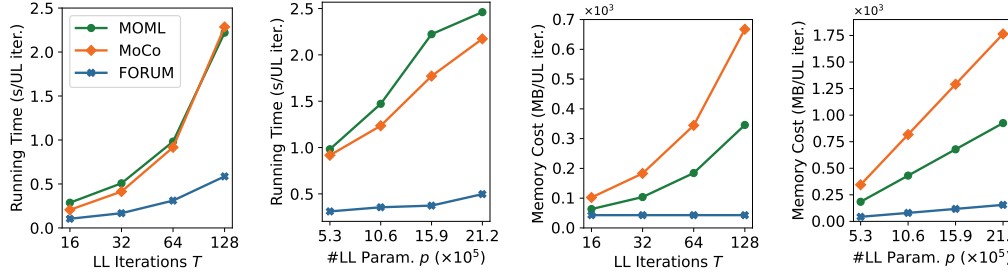

(a) Running time vs. $T$. (b) Running time vs. $p$. (c) Memory cost vs. $T$. (d) Memory cost vs. $p$.

Figure 1: Results of different methods on the multi-objective data hyper-cleaning problem. **(a)**: The running time per iteration varies over different LL update steps $T$. **(b)**: The running time per iteration varies over the different numbers of LL parameters $p$ with $T = 64$. **(c)**: The memory cost varies over different LL update steps $T$. **(d)**: The memory cost varies over the different numbers of LL parameters $p$ with $T = 64$.

### 5.2 MULTI-TASK LEARNING

**Setup.** Multi-Task Learning (MTL) (Caruana, 1997; Zhang & Yang, 2022) aims to train a single model to solve several tasks simultaneously. Following Ye et al. (2021), we aim to learn the loss weights to balance different tasks and improve the generalization performance by casting MTL as a MOBLO problem. Specifically, suppose there are $m$ tasks and the $i$-th task has its corresponding dataset $\mathcal{D}_i$ that contains a training set $\mathcal{D}_i^{\mathrm{tr}}$ and a validation set $\mathcal{D}_i^{\mathrm{val}}$. The MTL model is parameterized by $\omega$ and $\alpha \in \Delta^{m-1}$ denotes the loss weights for the $m$ tasks. Let $\mathcal{L}(\omega; \mathcal{D})$ represent the average loss of model $\omega$ on the dataset $\mathcal{D}$. The MOBLO formulation for MTL is as

$$\min_{\alpha,\omega} \left( \mathcal{L}(\omega; \mathcal{D}_1^{\mathrm{val}}), \cdots, \mathcal{L}(\omega; \mathcal{D}_m^{\mathrm{val}}) \right)^\top \ \text{s.t.} \ \omega \in \mathcal{S}(\alpha) = \arg\min_\omega \sum_{i=1}^m \alpha_i \mathcal{L}(\omega; \mathcal{D}_i^{\mathrm{tr}}).$$

We conduct experiments on three benchmark datasets among three different task categories, i.e., the Office-31 (Saenko et al., 2010) dataset for image classification, the NYUv2 (Silberman et al., 2012) dataset for scene understanding, and the QM9 dataset for molecular property prediction.

**Datasets.** (i) The Office-31 dataset (Saenko et al., 2010) includes images from three different sources: Amazon (A), digital SLR cameras (D), and Webcam (W). It contains 31 categories for

each source and a total of $4652$ labeled images. We use the data split in Lin et al. (2022): $60\%$ for training, $20\%$ for validation, and $20\%$ for testing. (ii) The NYUv2 dataset (Silberman et al., 2012), an indoor scene understanding dataset, has $795$ and $654$ images for training and testing, respectively. It has three tasks: 13-class semantic segmentation, depth estimation, and surface normal prediction. (iii) The QM9 dataset (Ramakrishnan et al., 2014), a molecular property prediction dataset. We use the commonly-used split as in Fey & Lenssen (2019); Navon et al. (2022): $110,000$ for training, $10,000$ for validation, and $10,000$ for testing. The QM9 dataset contains 11 tasks and each task is a regression task for one property. *Due to page limit, implementation details are put in Appendix D.2.*

**Baselines.** The proposed FORUM method is compared with: (i) *single-task learning* (STL) that trains each task independently; (ii) a comprehensive set of state-of-the-art *MTL methods*, including Equal Weighting (EW) (Zhang & Yang, 2022), UW (Kendall et al., 2018), PCGrad (Yu et al., 2020), GradDrop (Chen et al., 2020), GradVac (Wang et al., 2021), CAGrad (Liu et al., 2021a), Nash-MTL (Navon et al., 2022), and RLW (Lin et al., 2022); (iii) two *first-order BLO methods*: BVFIM (Liu et al., 2021c) and BOME (Liu et al., 2022a), where we simply transform MOBLO to BLO by aggregating multiple objectives in the UL subproblem with equal weights into a single objective so that we can apply those BLO methods to solve the MOBLO problem; (iv) two *MOBLO method*: MOML (Ye et al., 2021) and MoCo (Fernando et al., 2023).

Table 2: Classification accuracy (%) on the Office-31 dataset. Each experiment is repeated over 3 random seeds and the average is reported. The best results over baselines except STL are marked in **bold**.

| Methods | A | D | W | Avg | $\Delta_{\mathrm{P}}\uparrow$ |
|---|---|---|---|---|---|
| STL | 86.61 | 95.63 | 96.85 | 93.03 | 0.00 |
| *multi-task learning methods* | | | | | |
| EW | 83.53 | 97.27 | 96.85 | 92.55 | -0.61 |
| UW | 83.82 | 97.27 | 96.67 | 92.58 | -0.56 |
| PCGrad | 83.59 | 96.99 | 96.85 | 92.48 | -0.68 |
| GradDrop | 84.33 | 96.99 | 96.30 | 92.54 | -0.59 |
| GradVac | 83.76 | 97.27 | 96.67 | 92.57 | -0.58 |
| CAGrad | 83.65 | 95.63 | 96.85 | 92.04 | -1.13 |
| Nash-MTL | 85.01 | 97.54 | 97.41 | 93.32 | +0.24 |
| RLW | 83.82 | 96.99 | 96.85 | 92.55 | -0.59 |
| *first-order bi-level optimization methods* | | | | | |
| BVFIM | 84.84 | 96.99 | 97.78 | 93.21 | +0.11 |
| BOME | 85.53 | 96.72 | **98.15** | 93.47 | +0.41 |
| *multi-objective bi-level optimization methods* | | | | | |
| MOML | 84.67 | 96.72 | 96.85 | 92.75 | -0.36 |
| MoCo | 84.38 | 97.26 | 97.03 | 92.89 | -0.22 |
| FORUM (**ours**) | **85.64** | **98.63** | 97.96 | **94.07** | **+0.96** |

Table 3: Results on the NYUv2 dataset. Each experiment is repeated over 3 random seeds and the average is reported. The best results over baselines except STL are marked in **bold**. $\uparrow$ ($\downarrow$) indicates that the higher (lower) the result, the better the performance.

| Methods | Segmentation | | Depth | | Surface Normal Prediction | | | | | $\Delta_{\mathrm{P}}\uparrow$ |
|---|---|---|---|---|---|---|---|---|---|---|
| | | | | | Angle Distance | | Within $t°$ | | | |
| | mIoU↑ | PAcc↑ | AErr↓ | RErr↓ | Mean↓ | Median↓ | 11.25↑ | 22.5↑ | 30↑ | |
| STL | 53.50 | 75.39 | 0.3926 | 0.1605 | 21.9896 | 15.1641 | 39.04 | 65.00 | 75.16 | 0.00 |
| *multi-task learning methods* | | | | | | | | | | |
| EW | 53.93 | 75.53 | 0.3825 | 0.1577 | 23.5691 | 17.0149 | 35.04 | 60.99 | 72.05 | -1.78 |
| UW | **54.29** | 75.64 | 0.3815 | 0.1583 | 23.4805 | 16.9206 | 35.26 | 61.17 | 72.21 | -1.52 |
| PCGrad | 53.94 | 75.62 | 0.3804 | 0.1578 | 23.5226 | 16.9276 | 35.19 | 61.17 | 72.19 | -1.57 |
| GradDrop | 53.73 | 75.54 | 0.3837 | 0.1580 | 23.5392 | 16.9587 | 35.17 | 61.06 | 72.07 | -1.85 |
| GradVac | 54.21 | 75.67 | 0.3859 | 0.1583 | 23.5804 | 16.9055 | 35.34 | 61.15 | 72.10 | -1.75 |
| CAGrad | 53.97 | 75.54 | 0.3885 | 0.1588 | 22.4701 | 15.7139 | **37.77** | 63.82 | 74.30 | -0.27 |
| Nash-MTL | 53.41 | 74.95 | 0.3867 | 0.1612 | 22.5662 | 15.9365 | 37.30 | 63.40 | 74.09 | -1.01 |
| RLW | 54.13 | 75.72 | 0.3833 | 0.1590 | 23.2125 | 16.6166 | 35.88 | 61.84 | 72.74 | -1.27 |
| *first-order bi-level optimization methods* | | | | | | | | | | |
| BVFIM | 53.29 | 75.07 | 0.3981 | 0.1632 | 22.3552 | 15.9710 | 37.15 | 63.44 | 74.27 | -1.68 |
| BOME | 54.15 | **75.79** | 0.3831 | 0.1578 | 23.3378 | 16.8828 | 35.29 | 61.31 | 72.40 | -1.45 |
| *multi-objective bi-level optimization methods* | | | | | | | | | | |
| MOML | 53.59 | 75.48 | 0.3839 | 0.1577 | 23.1487 | 16.5319 | 36.06 | 62.05 | 72.89 | -1.26 |
| MoCo | 53.73 | 75.63 | 0.3838 | 0.1560 | 23.1922 | 16.5737 | 36.02 | 61.93 | 72.82 | -1.06 |
| FORUM (**ours**) | 54.04 | 75.64 | **0.3795** | **0.1555** | **22.1870** | **15.6815** | 37.71 | **64.04** | **74.67** | **+0.65** |

**Evaluation Metrics.** (i) For the Office-31 dataset, following Lin et al. (2022), we use classification accuracy as the evaluation metric for each task and the average accuracy as the overall metric. (ii) For the NYUv2 dataset, following Liu et al. (2019); Lin et al. (2022), we use the mean intersection

Table 4: Results on the QM9 dataset. Each experiment is repeated over 3 random seeds and the average is reported. ↑ (↓) indicates that the higher (lower) the result, the better the performance. The best results over baselines except STL are marked in **bold**.

| Methods | $\mu$ | $\alpha$ | $\epsilon_{\text{HOMO}}$ | $\epsilon_{\text{LUMO}}$ | $\langle R^2 \rangle$ | ZPVE | $U_0$ | $U$ | $H$ | $G$ | $c_v$ | $\Delta_{\text{p}}\uparrow$ |
|---|---|---|---|---|---|---|---|---|---|---|---|---|
| | | | | | | MAE↓ | | | | | | |
| STL | 0.062 | 0.192 | 58.82 | 51.95 | 0.529 | 4.52 | 63.69 | 60.83 | 68.33 | 60.31 | 0.069 | 0.00 |
| *multi-task learning methods* | | | | | | | | | | | | |
| EW | 0.096 | 0.286 | **67.46** | 82.80 | 4.655 | 12.4 | 128.3 | 128.8 | 129.2 | 125.6 | 0.116 | -146.3 |
| UW | 0.336 | 0.382 | 155.1 | 144.3 | **0.965** | **4.58** | **61.41** | **61.79** | **61.83** | **61.40** | 0.116 | -92.35 |
| PCGrad | 0.104 | 0.293 | 75.29 | 88.99 | 3.695 | 8.67 | 115.6 | 116.0 | 116.2 | 113.8 | 0.109 | -117.8 |
| GradDrop | 0.114 | 0.349 | 75.94 | 94.62 | 5.315 | 15.8 | 155.2 | 156.1 | 156.6 | 151.9 | 0.136 | -191.4 |
| GradVac | 0.100 | 0.299 | 68.94 | 84.14 | 4.833 | 12.5 | 127.3 | 127.8 | 128.1 | 124.7 | 0.117 | -150.7 |
| CAGrad | 0.107 | 0.296 | 75.43 | 88.59 | 2.944 | 6.12 | 93.09 | 93.68 | 93.85 | 92.32 | 0.106 | -87.25 |
| Nash-MTL | 0.115 | **0.263** | 85.54 | 86.62 | 2.549 | 5.85 | 83.49 | 83.88 | 84.05 | 82.96 | 0.097 | -73.92 |
| RLW | 0.112 | 0.331 | 74.59 | 90.48 | 6.015 | 15.6 | 156.0 | 156.8 | 157.3 | 151.6 | 0.133 | -200.9 |
| *first-order bi-level optimization methods* | | | | | | | | | | | | |
| BVFIM | 0.107 | 0.325 | 73.18 | 98.97 | 5.336 | 21.4 | 200.1 | 201.2 | 201.8 | 195.5 | 0.148 | -228.5 |
| BOME | 0.105 | 0.318 | 72.10 | 88.52 | 4.984 | 12.6 | 138.8 | 139.4 | 140.0 | 136.1 | 0.124 | -164.1 |
| *multi-objective bi-level optimization methods* | | | | | | | | | | | | |
| MOML | **0.083** | 0.347 | 74.87 | 80.57 | 3.813 | 8.64 | 191.9 | 192.6 | 192.8 | 188.9 | 0.135 | -165.1 |
| MoCo | 0.086 | 0.427 | 69.60 | **79.00** | 5.693 | 10.2 | 295.5 | 296.6 | 297.0 | 290.1 | 0.169 | -267.6 |
| FORUM (**ours**) | 0.104 | 0.266 | 85.37 | 82.15 | 2.126 | 6.49 | 96.97 | 97.53 | 97.69 | 95.88 | **0.097** | **-73.36** |

over union (MIoU) and the class-wise pixel accuracy (PAcc) for the semantic segmentation task, the relative error (RErr) and the absolute error (AErr) for the depth estimation task, and the mean and median angle error as well as the percentage of normals within $t°$ ($t = 11.25, 22.5, 30$) for the surface normal prediction task. (iii) For the QM9 dataset, following Fey & Lenssen (2019); Navon et al. (2022), we use mean absolute error (MAE) as the evaluation metric. (iv) Following Maninis et al. (2019); Lin et al. (2022), we use $\Delta_{\text{p}}$ as a metric to evaluate the overall performance on all the tasks. It is defined as the mean of the relative improvement of each task over the STL method, which is formulated as

$$\Delta_{\text{p}} = 100\% \times \frac{1}{m} \sum_{i=1}^{m} \frac{1}{N_i} \sum_{j=1}^{N_i} \frac{(-1)^{s_{i,j}} (M_{i,j} - M_{i,j}^{\text{STL}})}{M_{i,j}^{\text{STL}}},$$

where $N_i$ denotes the number of metrics for $i$-th task, $M_{i,j}$ denotes the performance of an MTL method for the $j$-th metric in the $i$-th task, $M_{i,j}^{\text{STL}}$ is defined in the same way for the STL method, and $s_{i,j}$ is set to 0 if a larger value represents better performance for the $j$-th metric in $i$-th task and otherwise $s_{i,j} = 1$.

**Results.** Table 2 shows the results on Office-31 dataset. The proposed FORUM method achieves the best performance in terms of average classification accuracy and $\Delta_{\text{p}}$. The results on NYUv2 dataset are shown in Table 3. As can be seen, only FORUM achieves better performance than STL in terms of $\Delta_{\text{p}}$. Moreover, FORUM performs well in the depth estimation and surface normal prediction tasks. Table 4 shows the results on QM9 dataset. FORUM again outperforms all the baselines in terms of $\Delta_{\text{p}}$. Those results consistently demonstrate FORUM achieves state-of-the-art performance and is more effective than previous MOBLO methods such as MOML and MoCo.

## 6 CONCLUSION

In this paper, we propose FORUM, an efficient fully first-order gradient-based method for solving the MOBLO problem. Specifically, we reformulate the MOBLO problem to a constrained MOO problem and we propose a novel multi-gradient aggregation method to solve it. Compared with the existing MOBLO methods, FORUM does not require any hypergradient computation and thus is efficient. Theoretically, we provide a complexity analysis to show the efficiency of the proposed method and a non-asymptotic convergence guarantee for FORUM. Moreover, empirical studies demonstrate the proposed FORUM method is effective and efficient.

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

# APPENDIX

## A  ADDITIONAL MATERIALS

### A.1  GRADIENT APPROXIMATION ERROR BOUND

The following lemma shows that the gradient approximation error of $\nabla_z \widetilde{q}(z)$ exponentially decays w.r.t. the LL iterations $T$.

**Lemma A.1.** *Under Assumption 4.2 and suppose the step size $\eta$ satisfies $\eta \leq \frac{2}{L_f + c}$, then we have*
$\|\nabla_z \widetilde{q}(z) - \nabla_z q(z)\| \leq L_f (1 - \frac{c\eta}{2})^T \|\tilde{\omega}^0 - \omega^*\|$.

*Proof.* According to Lemma 3 in Sow et al. (2022), as $\eta \leq \frac{2}{L_f + c}$, for a given $\alpha$, we have $\|\omega^T - \omega^*\| \leq (1 - \frac{c\eta}{2})^T \|\omega^0 - \omega^*\|$. Then for the approximated gradient $\nabla_\alpha f_z(\alpha, \tilde{\omega}^T)$, we have

$$\|\nabla_z \widetilde{q}(z) - \nabla_z q(z)\| = \|\nabla_\alpha f(\alpha, \omega^*) - \nabla_\alpha f(\alpha, \tilde{\omega}^T)\| \leq L_f \|\omega^* - \tilde{\omega}^T\|.$$

Then we reach the conclusion. $\qquad\square$

As $0 < \eta \leq \frac{2}{L_f + c}$, we have $0 < \frac{c\eta}{2} < 1$. Therefore, the gradient approximation error exponentially decays w.r.t. the LL iterations $T$ according to Lemma A.1.

### A.2  DUAL PROBLEM OF PROBLEM (7)

The sub-problem can be rewritten equivalently as the following differentiable quadratic optimization

$$d, \mu = \arg\min_{d,\mu} (\frac{1}{2} \|d\|^2 + \mu) \ \text{ s.t. } \ \langle \nabla \widetilde{q}(z_k), d \rangle \leq -\phi, \ \ \langle \nabla F_i(z_k), d \rangle \leq \mu \tag{13}$$

Therefore we have

$$L = \frac{1}{2} \|d\|^2 + \mu + \sum_{i=1}^{m} \lambda_i (\langle \nabla F_i(z_k), d \rangle - \mu) + \nu(\langle \nabla \widetilde{q}(z_k), d \rangle + \phi), \ \text{ s.t. } \ \sum_{i=1}^{m} \lambda_i = 1 \tag{14}$$

Differentiate with respect to $d$, and let $\nabla_d L = 0$ we have

$$d + \sum_{i=1}^{m} \lambda \nabla F_i(z_k) + \nu \nabla \widetilde{q}(z_k) = 0. \tag{15}$$

Therefore, the gradient $d = -(\sum_{i=1}^{m} \lambda \nabla F_i(z_k) + \nu \nabla \widetilde{q}(z_k))$. Substitute it to problem (13), we obtain that $\lambda$ and $\nu$ are the solution of

$$\min_{\lambda \in \Delta^{m-1}, \nu \geq 0} \frac{1}{2} \left\| \sum_{i=1}^{m} \lambda \nabla F_i(z_k) + \nu \nabla \widetilde{q}(z_k) \right\|^2 - \nu\phi. \tag{16}$$

For given $\lambda$, the above equation has a colsed form solution for $\nu$,

$$\nu(\lambda) = \max\left( \sum_{i=1}^{m} \lambda_i \pi_i(z), 0 \right), \ \text{ s.t. } \ \pi_i(z) = \frac{2\phi - \langle \nabla \widetilde{q}(z), \nabla F_i(z) \rangle}{\|\nabla \widetilde{q}(z)\|^2}. \tag{17}$$

Therefore, the optimization problem of $\lambda$ becomes the following quadratic programming problem.

$$\lambda = \arg\min_{\lambda \in \Delta^{m-1}} \left\| \sum_{i=1}^{m} \lambda_i \nabla F_i(z) + \nu(\lambda) \nabla \widetilde{q}(z) \right\|^2 - \nu(\lambda)\phi. \tag{18}$$

which fits the result in Section 3.2.

## A.3 Discussion on the Weights Sequence

For a standard non-convex MOO problem, i.e. $\min_z(F_1(z), F_2(z), \ldots, F_m(z))$. The goal is to find a Pareto stationary point $z^*$, i.e., exists a $\lambda \in \Delta^{m-1}$ satisfies $\sum_{i=1}^m \lambda_i \nabla F_i(z^*) = 0$. Suppose we use a gradient aggregated approach to update $z$, i.e., $z_{k+1} = z_k - \sum_{i=1}^m \mu \lambda_i^k \nabla F_i(z_k)$, where the gradient weights $\lambda$ can be obtained by any gradient-based MOO methods, such as MGDA (Désidéri, 2012) and CAGrad (Liu et al., 2021a). We now show that the convergence of reaching a stationary point can only be guaranteed when $\{\lambda^k\}_{k=1}^K$ is a convergent sequence.

Under Assumption 4.1, since all objective functions are $L_F$-smooth, we have

$$\sum_{i=1}^m \lambda_i^k (F_i(z_{k+1}) - F_i(z_k)) \leq -\mu \| \sum_{i=1}^m \lambda_i^k \nabla F_i(z_k) \|^2 + \frac{\sum_{i=1}^m \lambda_i^k L_F \mu^2}{2} \| \sum_{i=1}^m \lambda_i^k \nabla F_i(z_k) \|^2.$$

Let $\mu \leq \frac{1}{L_F}$ and sum the above inequality over $k = 0, 1, \ldots, K-1$ yields

$$\sum_{k=1}^{K-1} \frac{\mu}{2} \| \sum_{i=1}^m \lambda_i^k \nabla F_i(z_k) \|^2 \leq \sum_{k=0}^{K-1} \sum_{i=1}^m \lambda_i^k (F_i(z_{k+1}) - F_i(z_k)).$$

Therefore, we have

$$\frac{1}{K} \sum_{k=1}^{K-1} \| \sum_{i=1}^m \lambda_i^k \nabla F_i(z_k) \|^2 \leq \frac{\sum_{k=0}^{K-1} \sum_{i=1}^m (\lambda_i^k - \lambda_i^{k+1}) F(z_{k+1})}{\mu K} + \frac{\sum_{i=1}^m (\lambda_i^{K-1} F(z_K) - \lambda_i^0 F(z_0))}{\mu K}.$$

Suppose the value of the functions $F_i$ are bounded, by carefully choosing the step size $\mu$, i.e., $\mu = \mathcal{O}(K^{-1/2})$, the second term of the right side of the above inequality can maintain a $K^{-1/2}$ convergent rate. However, if $\|\lambda^k - \lambda^{k+1}\|$ does not converge to zero, the first term is of order $\mathcal{O}(\mu^{-1})$. Thus it can not converge to zero for both constant step size or decreased step size. Therefore, the $\{\lambda^k\}_{k=1}^K$ should be a convergent sequence to guarantee the convergence of the gradient aggregated-based method for non-convex MOO problems.

## B Convergence Analysis

### B.1 KKT Conditions

Consider a general constrained MOO problem with one constraint such as problem (3). The corresponding first-order Karush-Kuhn-Tucker (KKT) condition (Feng & Li, 2018) is said to hold at a feasible point $z^* \in \mathcal{Z}$ if there exist a vector $\lambda \in \Delta^{m-1}$ and $\nu \in \mathbb{R}_+$ such that the following three conditions hold,

$$\sum_{i=1}^m \lambda_i \nabla F_i(z^*) + \nu \nabla q(z^*) = 0, \ \ q(z^*) \leq 0, \text{and } \nu q(z^*) = 0. \tag{19}$$

Then $z^*$ is a local optimal point. The first condition is the stationarity condition, the second condition is the primal feasibility condition and the last condition is the complementary slackness condition. However, as we discussed in Section 3.1, since the constraint function $q(z)$ is ill-posed, the complementary slackness condition can not be satisfied (Liu et al., 2022a; Kwon et al., 2023). To ensure our algorithm converges to a weak stationarity point, we measure the convergence by the stationarity condition and the feasibility condition.

**Discussion on Pareto stationary.** The Pareto stationary is a first-order KKT stationary condition for the unconstrained MOO optimization problem. However, in this work, we reformulate MOBLO to an equivalent constrained MOO problem. Hence, the Pareto stationary cannot be used as a convergence criterion in our method. We measure the convergence by the local optimality condition of the constrained MOO problem, i.e., KKT stationary and feasibility conditions.

### B.2 Lemmas

We first provide the following lemmas for the LL subproblem under Assumption 4.2.

**Lemma B.1.** *Under Assumption 4.2, we have the following results.*

(a) $\|\nabla\widetilde{q}(z) - \nabla q(z)\| \leq L_f\|\tilde{\omega}^T - \omega^*(\alpha)\|$.

(b) *The function $\nabla q(z)$ is $L_q$-Lipschitz continuous w.r.t. $z$, where $L_q = L_f(2 + \frac{L_f}{c})$.*

(c) *If the step size of the LL subproblem satisfies $\eta \leq \frac{2}{L_f}$, then for $\tilde{\omega}^0 = \omega$ and $\tilde{\omega}^{T+1} = \tilde{\omega}^T - \eta\nabla_\omega q(\alpha, \tilde{\omega}^T)$, we have $q(\alpha, \tilde{\omega}^T) \leq \Gamma(T)q(\alpha, \omega)$, where $\Gamma(T)$ represents an exponential decay w.r.t. $T$.*

(d) $\|\nabla_z q(z)\|^2 \leq \frac{2L_q^2}{c}q(z)$

*Proof.* (a): We have $\|\nabla\widetilde{q}(z) - \nabla q(z)\| \leq \|\nabla_\alpha f(\alpha, \tilde{\omega}^T) - \nabla_\alpha f(\alpha, \omega^*)\| \leq L_f\|\tilde{\omega}^T - \omega^*(\alpha)\|$.

(b): This result can be directly obtained from Lemma 5 in Sow et al. (2022).

(c): Since $\nabla_\omega q(z) = \nabla_\omega f(z)$ and $\nabla_z f(z)$ is $L_f$-Lipschitz continuous w.r.t $\omega$, $\nabla_\omega q(z)$ is $L_f$-Lipschitz continuous w.r.t $\omega$. Then we have

$$q(\alpha, \tilde{\omega}^{T+1}) \leq q(\alpha, \tilde{\omega}^T) - (\eta - \frac{L_f\eta^2}{2})\|\nabla_\omega q(\alpha, \tilde{\omega}^T)\|^2$$

$$= q(\alpha, \tilde{\omega}^T) - (\eta - \frac{L_f\eta^2}{2})\|\nabla_\omega f(\alpha, \tilde{\omega}^T)\|^2$$

$$\leq (1 - c(2\eta - L_f\eta^2))q(\alpha, \tilde{\omega}^T),$$

where the second inequality is due to $\|\nabla_\omega f(\alpha, \tilde{\omega}^T)\|^2 \geq 2c(f(\alpha, \tilde{\omega}^T) - f(\alpha, \omega^*)) = 2cq(\alpha, \tilde{\omega}^T)$. Since $\tilde{\omega}^0 = \omega$, we have $q(\alpha, \tilde{\omega}^T) \leq (1 - c(2\eta - L_f\eta^2))^Tq(\alpha, \omega)$. If $\eta \leq \frac{2}{L_f}$, $2\eta - L_f\eta^2 \geq 0$. Let $\Gamma(T) = (1 - c(2\eta - L_f\eta^2))^T$, which decays exponentially w.r.t. $T$. Then we reach the conclusion.

(d): Since $\nabla q(\alpha, \omega^*(\alpha)) = 0$, we have $\|\nabla q(z)\|^2 = \|\nabla q(z) - \nabla q(\alpha, \omega^*(\alpha))\|^2 \leq L_q^2\|\omega - \omega^*(\alpha)\|^2$. Since $f(z)$ is $c$-strongly convex with respect to $\omega$, we have $\|\omega - \omega^*(\alpha)\|^2 \leq \frac{2}{c}(f(\alpha, \omega) - f(\alpha, \omega^*(\alpha)) = \frac{2q(z)}{c}$, then we reach the conclusion. $\qquad\square$

Then for the Algorithm 1, we have following result about the constraint function.

**Lemma B.2.** *Under Assumption 4.2, suppose the sequence $\{z_k\}_{k=0}^K$ generated by Algorithm 1 satisifes $q(z_k) \leq B$, where $B$ is a positive constant. Then there exists a constant $C > 0$, if $T \geq C$, the following results hold.*

(a) $q(z_k) \leq \Gamma_1(k)B + \mathcal{O}(\Gamma(T) + \mu)$, where $\Gamma_1(k)$ represents an exponential decay w.r.t $k$.

(b) $\|\nabla\widetilde{q}(z) - \nabla q(z)\| \leq L_f\sqrt{\frac{\Gamma(T)}{c^2}}\|\nabla q(\alpha, \omega)\|$.

(c) *There exists a positive constant $C_b < 1$, such that $\|\nabla\widetilde{q}(z)\| \geq C_b\|\nabla q(z)\|$.*

(d) $\sum_{k=0}^{K-1}\|\nabla\widetilde{q}(z_k)\|^2 = \mathcal{O}(\frac{1}{\mu} + K\Gamma(T) + K\mu)$.

*Proof.* (a): According to Lemma B.1 (d) and the boundedness assumption of $q(z_k)$, the gradient norm $\|\nabla q(z_k)\|$ is also bounded. Let $G(z_k) = \sum_{i=1}^m \tilde{\lambda}_i^k\nabla F_i(z_k)$, we have $d = -\mu(G(z_k) + \nu\nabla q(z_k))$ and $\nu = \max(\frac{\rho\|\nabla\widetilde{q}(z_k)\|^2 - \langle\nabla\widetilde{q}(z_k), G(z_k)\rangle}{\|\nabla\widetilde{q}(z_k)\|^2}, 0)$. Then we can applying Lemma B.1 to Lemma 10 in Liu et al. (2022a), we obtain that there exist a constant $C > 0$, if $T \geq C$, we have

$$q(\alpha_k, \omega_k) \leq \Gamma_1(k)q(\alpha_0, \omega_0) + \mathcal{O}(\Gamma(T) + \mu), \tag{20}$$

where $\Gamma_1(k) = (1 - \mu C_a)^k$ represents an exponential decay w.r.t $k$ and $C_a$ is a positive constant depending on $\eta$ and $c$. Then we reach the conclusion.

(b): We have $\|\nabla\widetilde{q}(z) - \nabla q(z)\| \leq L_f\|\tilde{\omega}^T - \omega^*(\alpha)\| \leq L_f\sqrt{\frac{2(f(\alpha,\tilde{\omega}^T) - f(\alpha,\omega^*(\alpha)))}{c}}$, where the first inequality is due to Lemma B.1 (a). Then we have

$$\|\nabla\widetilde{q}(z) - \nabla q(z)\| \leq L_f\sqrt{\frac{2\Gamma(T)q(z)}{c}} \leq L_f\sqrt{\frac{\Gamma(T)}{c^2}}\|\nabla q(\alpha,\omega)\|,$$

where the first inequality is due to Lemma B.1 (c) and the second inequality is due to the strongly convex assumption of $f(\alpha,\omega)$ w.r.t $\omega$ and $\|\nabla_\omega q(\alpha,\omega)\| \leq \|\nabla q(z)\|$.

(c): By the triangle inequality and Lemma B.2 (b), we obtain

$$\|\nabla\widetilde{q}(z)\| \geq \|\nabla q(z)\| - \|\nabla\widetilde{q}(z) - \nabla q(z)\| \geq (1 - L_f\sqrt{\frac{\Gamma(T)}{c^2}})\|\nabla q(z)\|.$$

Then there exists a positive constant $C > 0$ such that when $T \geq C$, we have $0 < L_f\sqrt{\frac{\Gamma(C)}{c^2}} < 1$. Let $C_b = 1 - L_f\sqrt{\frac{\Gamma(C)}{c^2}}$, then we have $\|\nabla\widetilde{q}(z)\| \geq C_b\|\nabla q(z)\|$ and $C_b < 1$.

(d): By the triangle inequality and Lemma B.2 (b), we obtain

$$\|\nabla\widetilde{q}(z)\| \leq \|\nabla\widetilde{q}(z) - \nabla q(z)\| + \|\nabla q(z)\| \leq (1 + L_f\sqrt{\frac{\Gamma(T)}{c^2}})\|\nabla q(z)\|.$$

By defining $C_e = \left(1 + L_f\sqrt{\frac{\Gamma(T)}{c^2}}\right)^2$, we have

$$\sum_{k=0}^{K-1}\|\nabla\widetilde{q}(z_k)\|^2 \leq \sum_{k=0}^{K-1}C_e\|\nabla q(z_k)\|^2$$
$$\leq \sum_{k=0}^{K-1}\frac{2L_q^2 C_e}{c}(\Gamma_1(k)B + \mathcal{O}(\Gamma(T) + \mu)),$$

where the second inequality is due to Lemma B.1 (d) and Lemma B.2 (a). Since $\sum_{k=0}^{K}(1 - \mu C_a)^k = \mathcal{O}(\frac{1}{\mu})$ and $C_e$ decays to 1 as $T \to +\infty$, we have $\sum_{k=0}^{K-1}\|\nabla\widetilde{q}(z_k)\|^2 = \mathcal{O}(\frac{1}{\mu} + K\Gamma(T) + K\mu)$. $\square$

### B.3 PROOF OF THE THEOREM 4.3

Since $F_i(z)$ is $L_F$-Lipschitz continuous, we have

$$\sum_{i=1}^{m}\tilde{\lambda}_i^k(F_i(z_{k+1}) - F_i(z_k)) \leq \mu\left\langle\sum_{i=1}^{m}\tilde{\lambda}_i^k\nabla F_i(z_k), d_k\right\rangle + \frac{\sum_{i=1}^{m}\tilde{\lambda}_i^k L_F\mu^2}{2}\|d_k\|^2$$
$$= \mu\left\langle-\nu_k\nabla\widetilde{q}(z_k) - d_k, d_k\right\rangle + \frac{L_F\mu^2}{2}\|d_k\|^2$$
$$= (\frac{L_F\mu^2}{2} - \mu)\|d_k\|^2 - \mu\nu_k\langle\nabla\widetilde{q}(z_k), d_k\rangle.$$

According to the complementary slackness condition of problem (7), We have $\nu_k(\langle\nabla\widetilde{q}(z_k), d_k\rangle + \phi_k) = 0$, where $\nu_k = \nu(\tilde{\lambda}^k)$. Therefore $-\nu_k\langle\nabla\widetilde{q}(z_k), d_k\rangle = \nu_k\frac{\rho}{2}\|\nabla\widetilde{q}(z_k)\|^2$. Then if $\mu \leq \frac{1}{L_F}$, we have

$$\sum_{i=1}^{m}\tilde{\lambda}_i^k(F_i(z_{k+1}) - F_i(z_k)) \leq -\frac{\mu}{2}\|d_k\|^2 + \frac{\rho\mu\nu_k}{2}\|\nabla\widetilde{q}(z_k)\|^2. \tag{21}$$

Let $G(z_k) = \sum_{i=1}^{m} \tilde{\lambda}_i^k \nabla F_i(z_k)$, we have $\nu_k \|\nabla \widetilde{q}(z_k)\|^2 \leq \rho \|\nabla \widetilde{q}(z_k)\|^2 - \langle G(z_k), \nabla \widetilde{q}(z_k) \rangle$. Summing the inequality (21) over $k = 0, 1, \ldots, K-1$ yields

$$
\sum_{k=0}^{K-1} \sum_{i=1}^{m} \tilde{\lambda}_i^k (F_i(z_{k+1}) - F_i(z_k)) \leq \sum_{k=0}^{K-1} (-\frac{\mu}{2}\|d_k\|^2 + \frac{\rho\mu\nu_k}{2}\|\nabla\widetilde{q}(z_k)\|^2)
$$

$$
\leq -\frac{\mu}{2}\sum_{k=0}^{K-1}\|d_k\|^2 + \sum_{k=0}^{K-1}\frac{\rho^2\mu}{2}\|\nabla\widetilde{q}(z_k)\|^2 - \sum_{k=0}^{K-1}\frac{\rho\mu}{2}\langle G(z_k), \nabla\widetilde{q}(z_k)\rangle
$$

$$
\leq -\frac{\mu}{2}\sum_{k=0}^{K-1}\|d_k\|^2 + \sum_{k=0}^{K-1}\frac{\rho^2\mu}{2}\|\nabla\widetilde{q}(z_k)\|^2 + \sum_{k=0}^{K-1}\frac{\rho\mu M}{2}\|\nabla\widetilde{q}(z_k)\|,
$$

where the last inequality is by Cauchy-Schwarz inequality and $\|G(z_k)\| \leq \|\nabla_z F_i(z_k)\| \leq M$. Therefore, we further have

$$
\sum_{k=0}^{K-1}\|d_k\|^2 \leq \frac{2\sum_{k=0}^{K-1}\sum_{i=1}^{m}\tilde{\lambda}_i^k(F_i(z_{k+1}) - F_i(z_k))}{\mu} + \sum_{k=0}^{K-1}\rho^2\|\nabla\widetilde{q}(z_k)\|^2 + \sum_{k=0}^{K-1}\rho M\|\nabla\widetilde{q}(z_k)\|.
$$

$$(22)$$

For the first term of the right-hand side of the inequality (22), we have

$$
\sum_{k=0}^{K-1}\sum_{i=1}^{m}\tilde{\lambda}_i^k(F_i(z_{k+1}) - F_i(z_k)) = \sum_{k=0}^{K-1}\sum_{i=1}^{m}(\tilde{\lambda}_i^k - \tilde{\lambda}_i^{k+1})F(z_{k+1}) + \sum_{i=1}^{m}(\tilde{\lambda}_i^{K-1}F(z_K) - \tilde{\lambda}_i^0 F(z_0))
$$

$$
\leq \sum_{k=0}^{K-1}\sum_{i=1}^{m}|\tilde{\lambda}_i^k - (1-\beta)\tilde{\lambda}_i^k - \beta\lambda_i^{k+1}|M + 2M.
$$

$$
\leq \sum_{k=0}^{K-1}\beta\sum_{i=1}^{m}|\tilde{\lambda}_i^k - \lambda_i^{k+1}|M + 2M.
$$

Since $\lambda \in \Delta^{m-1}$, we have $|\tilde{\lambda}_i^k - \lambda_i^{k+1}| \leq 2$. Then we have

$$
\sum_{k=0}^{K-1}\|d_k\|^2 \leq \frac{2mMK\beta + 2M}{\mu} + \sum_{k=0}^{K-1}\rho^2\|\nabla\widetilde{q}(z_k)\|^2 + \rho M\sqrt{K}\sqrt{\sum_{k=0}^{K-1}\|\nabla\widetilde{q}(z_k)\|^2}
$$

$$
= \mathcal{O}(\frac{K\beta}{\mu}) + \mathcal{O}(\frac{M}{\mu}) + \mathcal{O}(\frac{1}{\mu} + K\Gamma(T) + K\mu) + \mathcal{O}(\sqrt{\frac{K}{\mu}} + K\sqrt{\Gamma(T)} + K\sqrt{\mu})
$$

$$
= \mathcal{O}(\frac{K\beta}{\mu} + K\Gamma(T) + \sqrt{\frac{K}{\mu}} + K\sqrt{\mu}),
$$

where the first inequality is by Holder's inequality and the first equality is due to Lemma B.2 (d). For the measure of stationarity $\mathcal{K}(z_k)$, we obtain that

$$
\mathcal{K}(z_k) = \|G(z_k) + \nu_k\nabla q(z_k)\|^2 \leq 2\|d_k\|^2 + 2\|\nu_k(\nabla\widetilde{q}(z_k) - \nabla q(z_k))\|^2. \quad (23)
$$

According to B.1 (d), we have $\|\nabla q(z_k)\| \leq \sqrt{\frac{2BL_q^2}{c}}$. Then we obtain

$$
\|\nu_k(\nabla\widetilde{q}(z_k) - \nabla q(z_k))\| \leq \left|\rho - \frac{\langle G(z_k), \nabla\widetilde{q}(z_k)\rangle}{\|\nabla\widetilde{q}(z_k)\|^2}\right|\|\nabla\widetilde{q}(z_k) - \nabla q(z_k)\|
$$

$$
\leq \rho\|\nabla\widetilde{q}(z_k) - \nabla q(z_k)\| + |\langle G(z_k), \frac{\nabla\widetilde{q}(z_k)}{\|\nabla\widetilde{q}(z_k)\|}\rangle|\frac{\|\nabla\widetilde{q}(z_k) - \nabla q(z_k)\|}{\|\nabla\widetilde{q}(z_k)\|}
$$

$$
\leq L_f\sqrt{\frac{\Gamma(T)}{c^2}}(\rho\|\nabla q(z_k)\| + \frac{1}{C_b}|\langle G(z_k), \frac{\nabla\widetilde{q}(z_k)}{\|\nabla\widetilde{q}(z_k)\|}\rangle|)
$$

$$
\leq L_f\sqrt{\frac{\Gamma(T)}{c^2}}(\rho L_q\sqrt{\frac{2B}{c}} + \frac{M}{C_b}),
$$

where the third inequality is due to Lemma B.2 (b) and (c), and the last inequality is due to the Cauchy-Schwarz inequality. Therefore, we have $\|\nu_k(\nabla \widetilde{q}(z_k) - \nabla q(z_k))\|^2 = \mathcal{O}(\Gamma(T))$. Then we can get

$$
\begin{aligned}
\min_{k<K} \mathcal{K}(z_k) &\leq \frac{1}{K} \sum_{k=0}^{K-1} \mathcal{K}(z_k) \\
&= 2\mathcal{O}(\frac{K\beta}{\mu K} + \Gamma(T) + \sqrt{\frac{1}{\mu K}} + \sqrt{\mu}) + 2\mathcal{O}(\Gamma(T)) \\
&= \mathcal{O}(\frac{\beta}{\mu} + \Gamma(T) + \sqrt{\frac{1}{\mu K}} + \sqrt{\mu}),
\end{aligned}
$$

where the first equality is due to Eq. (23). According to Lemma B.2 (a), we obtain $q(z_k) = \mathcal{O}(\Gamma_1(k) + \Gamma(T) + \mu)$. Thus we get

$$
\max\left\{ \min_{k<K} \mathcal{K}(z_k), q(z_k) \right\} = \mathcal{O}\left( \sqrt{\mu} + \sqrt{\frac{1}{\mu K}} + \frac{\beta}{\mu} + \Gamma(T) \right).
$$

Let $\mu = \mathcal{O}(K^{-1/2})$ and $\beta = \mathcal{O}(K^{-3/4})$, we reach the conclusion.

## C  SYNTHETIC MOBLO

In this section, we use one synthetic MOBLO problem to illustrate the convergence of the proposed FORUM method. We first consider the following problem,

$$
\min_{\alpha \in \mathbb{R}, \omega \in \mathbb{R}^2} (\|\omega - (1,\alpha)\|_2^2, \|\omega - (2,\alpha)\|_2^2) \quad \text{s.t.} \ \omega \in \arg\min_{\omega \in \mathbb{R}^2} \|\omega - \alpha\|_2^2, \tag{24}
$$

where $(\cdot, \cdot)$ denotes a two-dimensional vector and $\omega = (\omega_1, \omega_2)$. Problem (24) satisfies all the assumptions required in Section 4.2. By simple calculation, we can find that the optimal solution set of problem (24) is $\mathcal{P} = \{(\alpha, \omega) \mid \alpha = \omega_1 = \omega_2 = c, c \in [1, 2]\}$.

We apply GD optimizer for both UL and LL subproblems and the step sizes are set to $\mu = 0.3$ and $\eta = 0.05$ for all methods. We run $50$ LL iterations to ensure that for a given $\alpha$, they all reach the minimum point for the LL subproblem. For FORUM, we set $\rho = 0.3$ and $\beta_k = (k+1)^{-3/4}$. The result is evaluated by calculating the Euclidean distance between solution $z$ and the optimal solution set $\mathcal{P}$, which is denoted by $\mathcal{E} = \text{dist}(z, \mathcal{P})$.

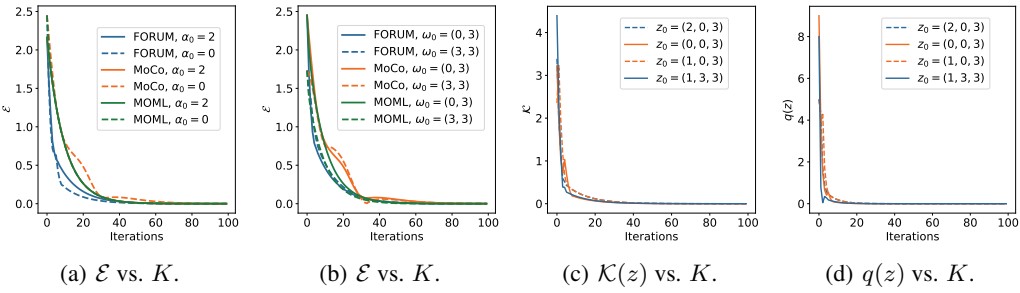

(a) $\mathcal{E}$ vs. $K$.  (b) $\mathcal{E}$ vs. $K$.  (c) $\mathcal{K}(z)$ vs. $K$.  (d) $q(z)$ vs. $K$.

Figure 2: Results on the problem (24) with different initialization points. **(a)**: Fix $\omega_0 = (0,3)$ and vary $\alpha_0 = 0, 2$. The optimality gap $\mathcal{E}$ curves. **(b)**: Fix $\alpha_0 = 2$ and vary $\omega_0 = (0,3), (3,3)$. The optimality gap $\mathcal{E}$ curves. **(c)**: The stationarity gap $\mathcal{K}(z)$ curves. **(d)**: The value of the constraint function $q(z)$ curves.

Figures 2(a) and 2(b) show the numerical results of the MOML, MoCo, and FORUM methods with different initializations. It can be observed that the proposed FORUM method can achieve an optimal solution in all the settings, i.e., $\mathcal{E} \to 0$, and different initializations only slightly affect the convergence speed. Figures 2(c) and 2(d) show that both $\mathcal{K}(z)$ and $q(z)$ converge to zero in all the settings. Thus FORUM solves the corresponding constrained optimization problem. This result demonstrates our convergence result in Section 4.2.

# D    IMPLEMENTATION DETAILS FOR SECTION 5

## D.1    DATA HYPER-CLEANING

The same configuration is used for both the MOML, MoCo, and FORUM methods. Specifically, the hard-parameter sharing architecture (Caruana, 1993) is used, where the bottom layers are shared among all datasets and each dataset has its specific head layers. The shared module contains two linear layers with input size, hidden size, and output size of $784$, $512$, and $256$. Each layer adopts a ReLU activation function. Each dataset has a specific linear layer with an output size of $10$. The batch size is set to $100$. For the LL subproblem, the SGD optimizer with a learning rate $\eta = 0.3$ is used for updating $T = 16, 32, 64, 128$ iterations. For the UL subproblem, the total number of UL iterations $K$ is set to 1200, and an SGD optimizer with the learning rate as $10$ is used for updating weight $\alpha$ while another SGD optimizer with the learning rate as $0.3$ is used for updating model parameters $\omega$. We set $\rho = 0.5$ and $\beta_k = (k+1)^{-\frac{3}{4}}$ for FORUM.

For Figures 1(b) and 1(d), we increase the number of LL parameters $p$ by adding some linear layers with the hidden size of $512$ into the shared module. We use the build-in function torch.cuda.max_memory_allocated in PyTorch (Paszke et al., 2019) to compute the GPU memory cost for Figures 1(c) and 1(d).

## D.2    MULTI-TASK LEARNING

All methods are implemented based on the open-source LibMTL library (Lin & Zhang, 2023). For the proposed FORUM method, we set $\rho = 0.1$, $\beta_k = (k+1)^{-\frac{3}{4}}$, use a SGD optimizer with a learning rate $\eta = 0.1$ to update $T = 5$ iterations in the LL subproblem, and use an Adam optimizer (Kingma & Ba, 2015) with the learning rate $10^{-3}$ to update the loss weight $\alpha$ in the UL subproblem.

**Office-31.** Following Lin et al. (2022), the ResNet-18 network pre-trained on the ImageNet dataset is used as a shared backbone among tasks and a fully connected layer is applied as a task-specific output layer for each task. All the input images are resized to $224 \times 224$. The batch size is set to $64$. The cross-entropy loss is used for all tasks in both datasets. The total number of UL epochs $K$ is set to 100. An Adam optimizer (Kingma & Ba, 2015) with the learning rate as $10^{-4}$ and the weight decay as $10^{-5}$ is used for updating model parameters $\omega$ in the UL subproblem.

**NYUv2.** Following Lin et al. (2022), we use the DeepLabV3+ architecture (Chen et al., 2018), which contains a ResNet-50 network with dilated convolutions as the shared encoder among all tasks and three Atrous Spatial Pyramid Pooling (ASPP) (Chen et al., 2018) modules as task-specific heads for each task. All input images are resized to $288 \times 384$. The batch size is set to $8$. The cross-entropy loss, $L_1$ loss, and cosine loss are used as the loss function of the semantic segmentation, depth estimation, and surface normal prediction tasks, respectively. The total number of UL epochs $K$ is set to 200. An Adam optimizer (Kingma & Ba, 2015) with the learning rate as $10^{-4}$ and the weight decay as $10^{-5}$ is used for updating model parameters $\omega$ in the UL subproblem. The learning rate of $\omega$ is halved after 100 epochs.

**QM9.** Following Fey & Lenssen (2019); Navon et al. (2022), we use a graph neural network (Gilmer et al., 2017) as the shared encoder, and a linear layer as the task-specific head. The targets of each task are normalized to have zero mean and unit standard deviation. The batch size is set to $128$. Following Fey & Lenssen (2019); Navon et al. (2022), we use mean squared error (MSE) as the loss function. The total number of UL epochs $K$ is set to 300. An Adam optimizer (Kingma & Ba, 2015) with the learning rate as $0.001$ is used for updating model parameters $\omega$ in the UL subproblem. A ReduceLROnPlateau scheduler (Paszke et al., 2019) is used to reduce the learning rate of $\omega$ once $\Delta_{\mathrm{p}}$ on the validation dataset stops improving.

For the BOME, BVFIM, MOML, and MoCo methods, we use a similar configuration to the proposed FORUM method and perform a grid search for hyperparameters of each method. Specifically, we search LL learning rate $\eta$ over $\{0.05, 0.1, 0.5\}$ for both four methods, search $\rho$ over $\{0.1, 0.5, 0.9\}$ for BOME, search $\beta$ over $\{0.05, 0.1, 0.5, 1\}$ for BVFIM, and set $T = 1$ for MOML and MoCo and $T = 5$ for BOME and BVFIM.

# E    EFFECTS OF $\eta$ AND $\rho$

We study the effects of hyperparameters $\eta$ and $\rho$ in the multi-objective data hyper-cleaning problem. The results are shown in Table 5. FORUM is insensitive with $\eta$ and a large $\rho$ (e.g, $\rho = 0.5, 0.7, 0.9$). Besides, FORUM with a positive $\rho$ performs better than $\rho = 0$, which shows the effectiveness of $\phi_k$ introduced in Section 3.2.

Table 5: Effects of $\eta$ and $\rho$ in the multi-objective data hyper-cleaning problem. The average accuracy on MNIST and FashionMNIST is reported.

| $\eta$ | $\rho$ | $T = 16$ | $T = 32$ | $T = 64$ | $T = 128$ |
|---|---|---|---|---|---|
|  | 0 | 83.94 | 84.07 | 84.30 | 83.94 |
|  | 0.1 | 85.44 | 85.52 | 85.50 | 85.32 |
| 0.3 | 0.3 | 85.99 | 86.18 | 85.92 | 85.96 |
|  | 0.5 | 86.58 | 86.38 | 86.44 | 86.10 |
|  | 0.7 | 86.53 | 86.64 | 86.66 | 86.48 |
|  | 0.9 | 86.91 | 86.61 | 86.51 | 86.65 |
| 0.1 |  | 86.57 | 86.61 | 86.16 | 86.24 |
| 0.3 | 0.5 | 86.58 | 86.38 | 86.44 | 86.10 |
| 0.5 |  | 86.52 | 86.57 | 86.20 | 86.31 |

# F    COMPARISON OF DIFFERENT MOBLO METHODS

Table 6 shows the convergence and complexity analysis for the proposed FORUM method and two MOBLO baselines (i.e., MOML and MoCo). As can be seen, FORUM has lower computational and memory costs compared with previous MOBLO methods.

Table 6: Comparison of convergence result and complexity analysis for different MOBLO methods.

| Method | Convergence analysis | Computational cost | Space cost |
|---|---|---|---|
| MOML | asymptotic | $\mathcal{O}(mp(n + p)T)$ | $\mathcal{O}(mn + mpT)$ |
| MoCo | non-asymptotic | $\mathcal{O}(mp(n + p)T)$ | $\mathcal{O}(2mn + mpT)$ |
| FORUM (**ours**) | non-asymptotic | $\mathcal{O}(mn + p(m + T))$ | $\mathcal{O}(mn + mp)$ |

