# OpenReview forum: "A First-Order Multi-Gradient Algorithm for Multi-Objective Bi-Level Optimization"
_ICLR.cc/2024/Conference — Submitted to ICLR 2024_

### Official Review · Reviewer_nzzn · 2023-10-25

**Soundness:** 3 good
**Presentation:** 3 good
**Contribution:** 3 good
**Rating:** 6
**Confidence:** 3

**Summary:**

In this work, the Multi-Objective Bi-Level Optimization (MOBLO) is studied, and an efficient first-order multi-gradient method for MOBLO, called FORUM, is proposed. The proposed method first reformulates MOBLO as an equivalent constrained multi-objective problem, then a novel multi-gradient aggregation method to solve the constrained multi-objective problem.

**Strengths:**

1. The proposed method combines the value-function-based approach and multi-gradient method, which is novel in the multiobjective bilevel optimization problems.

2. The writing of this work is good, and the logic of the proposed method is clear.

**Weaknesses:**

I believe this work is solid and good, however, I have some concerns as follows.

1. In the proposed method, an additional optimization problem is required to solve every iteration, i.e., Eq. (11). Thus the proposed method seems inefficient since it is a nested-loop algorithm.

2. I suggest the authors add a table to compare the differences between the proposed methods and existing MOBLO methods (i.e., MOML and MoCo) to clearly show the advantages of the proposed method.

**Questions:**

See Weaknesses above.

---

> ### Author Response · Authors · 2023-11-20
> **Reply to Reviewer nzzn**
>
> Thank you for your thoughtful review and constructive comments.
>
> ---
>
> >Q1. "In the proposed method, an additional optimization problem is required to solve every iteration, i.e., Eq. (11). Thus the proposed method seems inefficient since it is a nested-loop algorithm.''
>
> **A1**: (1) As discussed in the paragraph after problem (11), problem (11) is a quadratic programming problem and its first term of the objective function can be simplified to $R^\top \Lambda^\top \Lambda R$, where $R = (\lambda_1, \ldots , \lambda_m, \gamma)^\top$ and $\Lambda = (\nabla F_1, \ldots ,\nabla F_m,\nabla \widetilde{q})$. Note that the dimension of the matrix $\Lambda^\top \Lambda$ is $(m+1)\times(m+1)$, which is independent with the dimension of $z$. As the number of UL objectives $m$ is usually small, problem (11) can be solved efficiently.
>
> (2) We have provided the complexity analysis for the proposed FORUM method and two MOBLO baselines (i.e., MOML and MoCo) in Section 4.1, and showed their running time in Figure 1. According to the results, FORUM is more efficient than MOML and MoCo.
>
> ---
>
> >Q2. "I suggest the authors add a table to compare the differences between the proposed methods and existing MOBLO methods (i.e., MOML and MoCo) to clearly show the advantages of the proposed method.''
>
> **A2**: Thanks for your suggestion. In the following table where $n$, $p$, $m$ and $T$ denote the dimension of variable $\alpha$, the dimension of variable $\omega$, the number of objectives, and the number of LL iterations, respectively, we show the differences between the proposed methods and other gradient-based MOBLO methods (i.e. MOML and MoCo).
>
> \begin{array}{cccc}
>     \hline
>     \text{Method} & \text{Convergence analysis} & \text{Computational cost} & \text{Space cost} \newline
>     \hline
>     \text{MOML} & \text{asymptotic} & \mathcal{O}(mp(n+p)T) &  \mathcal{O}(mn+mpT)\newline
>     \text{MoCo} & \text{non-asymptotic}  & \mathcal{O}(mp(n+p)T) & \mathcal{O}(2mn+mpT)\newline
>     \text{FORUM} & \text{non-asymptotic} & \mathcal{O}(mn+p(m+T)) & \mathcal{O}(mn+mp)\newline
>     \hline
> \end{array}
>
> The main advantage of our method is that it has lower computational and memory costs. Our method also has better experimental results in practical problems, as observed in our experiments on multi-objective data hyper-cleaning and multi-task learning settings.
>
> We have added this table in Section F of the updated paper.

---

> > ### Comment · Reviewer_nzzn · 2023-11-22
> >
> > Thanks for responses. My concerns have been addressed and I'm happy to maintain the score as 6.

---

> > > ### Author Response · Authors · 2023-11-22
> > >
> > > Thanks for your reply. We are glad that our reply addressed your concerns.

---

### Official Review · Reviewer_6mPv · 2023-10-25

**Soundness:** 2 fair
**Presentation:** 3 good
**Contribution:** 2 fair
**Rating:** 5
**Confidence:** 4

**Summary:**

This paper introduces a first-order multi-gradient method tailored for the multi-objective bi-level optimization (MOBLO) problem. More precisely, the authors first reformulate the MOBLO problem as an equivalent single-level constrained multi-objective optimization problem using a value-function-based approach. Subsequently, they integrate the BOME method, designed for single-objective bi-level optimization, with the MGDA for multi-objective optimization (MOO) to address this equivalent problem. The authors provide convergence analysis and numerical results.

**Strengths:**

The topic of multi-objective bi-level optimization is important.

Convergence analysis is provided for the proposed method under some assumptions, e.g., the lower level problem is strongly convex.

Numerical validation is presented for the proposed method.

**Weaknesses:**

1. The proposed algorithm is a straightforward combination of two existing methods, and the accompanying analysis appears rather standard. Consequently, the technical innovation compared to prior work upon which this study is based is limited.

2. The convergence result of the proposed method lacks persuasiveness. As pointed out by the authors, the constraint $q(z) \le 0$ in the reformulated problem (3) is ill-posed, rendering the KKT stationary condition not a necessary condition for problem (3) solutions. Therefore, the utilization of the $\mathcal{K}(z)$ measure for the convergence of the proposed method in the main convergence theorem (Theorem 4.3) is inappropriate. In contrast, the MoCo method by Fernando et al. (2023) employs hyper-gradients under the same strong convexity in the LL problem to characterize convergence and establish convergence to Pareto stationarity.

3. The analysis of the proposed algorithm is confined to a deterministic setting, which may restrict its applicability given that the motivating applications are mostly in the stochastic setting.

4. The strong convexity of the LL problem appears to play a pivotal role in the convergence analysis, which, in turn, constrains the applicability of the proposed method. Notably, the MOML method introduced by Ye et al. (2022) does not necessitate such an assumption.

**Questions:**

My questions are listed in the “Weaknesses” part.

---

> ### Author Response · Authors · 2023-11-20
> **Reply to Reviewer 6mPv**
>
> Thank you for your thoughtful review and constructive comments.
>
> ---
>
> >Q1. "The proposed algorithm is a straightforward combination of two existing methods, and the accompanying analysis appears rather standard. Consequently, the technical innovation compared to prior work upon which this study is based is limited.''
>
> **A1**: The FORUM method is a novel approach for effectively solving the challenging Multi-Objective Bi-Level Optimization (MOBLO) problem. Unlike previous value-function-based approaches, our method specifically addresses the MOBLO problem for the first time. We further propose a novel max-min optimization problem to determine the aggregated weights and provide an update scheme with theoretical convergence analysis. Therefore, the proposed algorithm is not a straightforward combination of existing methods.
>
> ---
>
> >Q2. "The convergence result of the proposed method lacks persuasiveness. As pointed out by the authors, the constraint $q(z)\le 0$ in the reformulated problem (3) is ill-posed, rendering the KKT stationary condition not a necessary condition for problem (3) solutions. Therefore, the utilization of the $\mathcal{K}(z)$ measure for the convergence of the proposed method in the main convergence theorem (Theorem 4.3) is inappropriate. In contrast, the MoCo method by Fernando et al. (2023) employs hyper-gradients under the same strong convexity in the LL problem to characterize convergence and establish convergence to Pareto stationarity.''
>
> **A2**: The convergence of the proposed method can be measured by $\mathcal{K}(z)$ and $q(z)$. For a constrained optimization problem with a specific structure (i.e., $q(z)\le 0$ does not have an interior point), the convergence can be measured by the problem's stationarity condition and feasibility condition. This measure of convergence has been used for the value-function-based bi-level optimization method [1]. In our case, the condition $\sum_{i=1}^{m}\lambda_i \nabla F_i(z^*) + \nu \nabla q(z^*) =0$ indicates the local optimality of $F(z)$ as it is the KKT stationarity condition for the constrained multi-objective optimization problem. When $q(z^*)=0$, it indicates that $\omega^*$ is a minimum point of $f$, i.e., $\omega^* = \text{argmin}_{\omega} f(\alpha,\omega)$, ensuring the satisfaction of the feasibility condition $q(z^*)\le 0$. Therefore, $\mathcal{K}(z)$ and $q(z)$ can be used to measure the convergence of problem (3) and the original MOBLO problem, since these two problems are equivalent.
>
> MoCo uses a different reformulation strategy to simplify the original MOBLO problem into an unconstrained MOO problem. Therefore, they can measure the convergence by the Pareto stationarity. However, this approach employs hyper-gradients, which are computationally inefficient, as evidenced by the complexity analysis in Section 4.1 and experimental results in Figure 1. In our work, we focus on studying an efficient method to solve the MOBLO problem, so we do not apply their reformulation method.
>
> ---
>
> >Q3. "The analysis of the proposed algorithm is confined to a deterministic setting, which may restrict its applicability given that the motivating applications are mostly in the stochastic setting.''
>
> **A3**: Though the proposed method and its convergence analysis are based on the deterministic gradient descent, our experimental results demonstrate it is also suitable for stochastic gradient descent in practice and achieves state-of-the-art performance on two learning problems, i.e., data hyper-cleaning and multi-task learning.
>
> ---
>
> >Q4. "The strong convexity of the LL problem appears to play a pivotal role in the convergence analysis, which, in turn, constrains the applicability of the proposed method. Notably, the MOML method introduced by Ye et al. (2022) does not necessitate such an assumption.''
>
> **A4**: MOML does not need such an assumption since it only provides the asymptotic convergence analysis, i.e., the Pareto optimal set of $F(\alpha,\omega^T(\alpha))$ will converge to the set of $F(\alpha,\omega^*(\alpha))$. However, in this paper, we provide the non-asymptotic convergence analysis, where we need to reach the solution of the Pareto optimal set of $F(\alpha,\omega^T(\alpha))$, thus, the strong convexity assumption of the LL problem is needed. Similarly, MoCo also needs the strong convexity assumption of the LL problem for the non-asymptotic convergence analysis.
>
> ---
>
> **References**
>
> [1] Chengyue Gong, Xingchao Liu, and Qiang Liu. Automatic and harmless regularization with con-
> strained and lexicographic optimization: A dynamic barrier approach. In Neural Information Processing Systems, 2021.

---

> > ### Comment · Reviewer_6mPv · 2023-12-04
> >
> > Thanks for the response.

---

### Official Review · Reviewer_xYWk · 2023-10-30

**Soundness:** 2 fair
**Presentation:** 2 fair
**Contribution:** 2 fair
**Rating:** 3
**Confidence:** 3

**Summary:**

This work studied a multi-objective optimization problem where each objective has a bilevel optimization structure. Each upper-level objective is evaluated at the minimum of the same lower-level problem. The authors utilize the idea from value-function-based method to solving the bilevel problem as well as a momentum update idea in Zhou et al., 2022 in the multi-objective algorithm. A convergence analysis is provided for the derived algorithm. Several experiments on hyper-cleaning, multi-task learning over Office-31, NYUv2, QM9 datasets are provided.

**Strengths:**

1.	Overall, the studied problem is timely. Bilevel optimization and multi-objective optimization have wide applications in practice. This work uses a first-order idea from bilevel literature in MOO, and seems to work well in experiments.

2.	Experiments seem to support that the proposed method can work well in some datasets.

**Weaknesses:**

1.	However, I have quite a few concerns regarding the analysis and the novelty. First, to deal with the reformulated constraint in the value-function based method, the authors use a dot product between $d$ and the gradient of the constraint and make it less or equal to $-\phi_k$. However, how to pick this $\phi_k$ in practice and in theory? Also, the final convergence criterion seems questionable. For example, the authors use the measure of KKT stationary condition to as convergence criterion. How does this condition correlate with Parato stationarity? How fast does the parameter $v_k$ decrease to 0 in this criterion? All these questions are not well explained in this work.
2.	The algorithm has some unclear parts. For example, the authors use the idea of momentum update on $\lambda$ update. This step is originally proposed in Zhou et al., 2022 (the authors should mention about this). However, there is characterization on the distance between the true variable $\lambda_k$ and the surrogate $\titilde \lambda_k$? This is important, because what you need to use is $\lambda_k$ rather than surrogate $\titilde \lambda_k$ in the algorithm.
3.	The algorithm is deterministic without data sampling, but in the experiments it seems data sampling is used. I am wondering if it is possible to extend the algorithm and analysis to the more practice stochastic setting? If not, what are the challenges? Some recent progresses on stochastic MOO may be helpful here (some of them are missing in this work).

[1] Suyun Liu and Luis Nunes Vicente. The stochastic multi-gradient algorithm for multi-objective optimization and its application to supervised machine learning.

[2] Lisha Chen, Heshan Fernando, Yiming Ying, and Tianyi Chen. Three-way trade-off in multi-objective learning: Optimization, generalization and conflict-avoidance.

[3] Heshan Devaka Fernando, Han Shen, Miao Liu, Subhajit Chaudhury, Keerthiram Murugesan, and Tianyi Chen. Mitigating gradient bias in multi-objective learning: A provably convergent approach.

[4] Peiyao Xiao, Hao Ban, and Kaiyi Ji. Direction-oriented Multi-objective Learning: Simple and Provable Stochastic Algorithms.

4.	The analysis assumes the upper-level function $F_i$ is bounded. However, this has not been made in the aforementioned [1,2,3,4] works. More clarifications should be provided. If there are any special challenges making this assumption necessary?  Also, the assumption $q(z_k)<B$ is a strong assumption that has not been made in previous bilevel and MOO literatures, because bounded function value and strong-convexity cannot be made simultaneously.

**Questions:**

Overall, this work studied an interesting and important problem. However, it has quite a few questions and problems to be solved. However, I am open to increase my score given the authors’ response. See my questions in the weakness part.

---

> ### Author Response · Authors · 2023-11-20
> **Reply to Rreviewer J6Rt (1/3)**
>
> Thank you for your thoughtful review and constructive comments.
>
> ---
>
> >Q1. "However, I have quite a few concerns regarding the analysis and the novelty. First, to deal with the reformulated constraint in the value-function based method, the authors use a dot product between $d$ and the gradient of the constraint and make it less or equal to $-\phi_k$. However, how to pick this $\phi_k$ in practice and in theory? Also, the final convergence criterion seems questionable. For example, the authors use the measure of KKT stationary condition to as convergence criterion. How does this condition correlate with Parato stationarity? How fast does the parameter $v_k$ decrease to 0 in this criterion? All these questions are not well explained in this work.''
>
> **A1**: (i) We have discussed how to pick this $\phi_k$ in Section 3.2 in the submission. We set $\phi_k=\frac{\rho}{2}||\nabla \widetilde{q}(z_k)||^2$ in practice and in theoretical analysis, where $\rho$ is a positive constant.
> Then when $\phi_k>0$, it means that $||\nabla \tilde{q}(z)||\not=0$ and $\widetilde{q}(z)$ should be further optimized, and $\langle \nabla \widetilde{q}(z_k),-d\rangle \ge \phi_k > 0$ can enforce $\widetilde{q}(z)$ to decrease. When $\phi_k$ equals $0$, it indicates that the optimum of $\widetilde{q}(z)$ is reached and $\langle \nabla \widetilde{q}(z_k),-d\rangle \ge \phi_k = 0$ also holds. Thus, the dynamic $\phi_k$ can ensure $d_k$ to iteratively decrease $\widetilde{q}(z)$ such that the constraint $\widetilde{q}(z)\le0$ is satisfied.
>
> (ii) It is worth noting that the FORUM algorithm is proposed for MOBLO rather than MOO. The Pareto stationary is a first-order KKT stationary condition for the **unconstrained** MOO optimization problem. However, we reformulate MOBLO to an equivalent **constrained** MOO problem. Hence, the Pareto stationary cannot be used as a convergence criterion in our method. We measure the convergence by the local optimality condition of the constrained MOO problem, i.e., KKT stationary and feasibility conditions.
>
> We have added this discussion in Section B.1 of the updated paper.
>
> (iii) The KKT stationary condition only requires $\nu_k \ge 0$ and $\mathcal{K}(z_k) =0$. Since $\nabla q(z_k)$ will converge to 0 as $q(z_k)\to 0$, $\nu_k$ is not necessary to decrease to 0.
>
> ---
>
> >Q2. "The algorithm has some unclear parts. For example, the authors use the idea of momentum update on $\lambda$ update. This step is originally proposed in Zhou et al., 2022 (the authors should mention about this). However, there is characterization on the distance between the true variable $\lambda_k$ and the surrogate $\tilde{\lambda}_k$? This is important, because what you need to use is $\lambda_k$ rather than surrogate $\tilde{\lambda}_k$ in the algorithm.''
>
> **A2**: Thanks for your suggestion. We have mentioned this in Section 3.2 in the updated paper. The distance between $\lambda^k$ and $\tilde{\lambda}^k$  can be expressed as $||\lambda^k- \tilde{\lambda}^k|| = (1-\beta_k) ||\lambda^k- \tilde{\lambda}^{k-1} ||$. As a result, the distance is upper-bounded by $\sqrt{2}$. In the initial stages, when $k$ is small, $\beta_k$ approximates $1$, leading to a small distance. This allows the algorithm to retain the advantage of maximizing the minimum decrease across all upper-level objectives. As $\beta_k \to 0$ and $\tilde{\lambda}^k$ converge to a limit point, the distance increases. So in the later stage, the FORUM method performs similarly to the fixed linear scalarization for the UL subproblem, ensuring convergence.
>
> ---

---

> ### Author Response · Authors · 2023-11-20
> **Reply to Rreviewer J6Rt (2/3)**
>
> >Q3. "The algorithm is deterministic without data sampling, but in the experiments it seems data sampling is used. I am wondering if it is possible to extend the algorithm and analysis to the more practice stochastic setting? If not, what are the challenges? Some recent progresses on stochastic MOO may be helpful here (some of them are missing in this work)."
>
> **A3**: (i) Extending the analysis of our algorithm to the stochastic setting might be challenging. We think there are two main technical difficulties. Firstly, for stochastic gradient descent, the lower-level optimization can not maintain an exponential convergence rate, so Lemma B.1. (c) no longer holds. This will affect our convergence proof. Secondly, the stochastic setting introduces an approximation error term $||\nabla_z q(z) - \nabla_z q(z;\xi)||$, where $\nabla_z q(z;\xi)$ represents the stochastic gradient. Though we can make a boundedness assumption for this error, it will still introduce an additional constant term for our current boundedness result such as Lemma B.2. (b), and it is hard to cancel this term for our nested algorithm. It is interesting to analyze the effect of the stochastic gradient approximation on our algorithm theoretically, and we regarded it as a future work.
>
> (ii) Though the proposed method and its convergence analysis are based on deterministic gradient descent, our experimental results demonstrate its suitability for stochastic gradient descent in practice. Our method achieves state-of-the-art performance on two learning problems, i.e., data hyper-cleaning and multi-task learning.
>
> ---

---

> ### Author Response · Authors · 2023-11-20
> **Reply to Rreviewer J6Rt (3/3)**
>
> >Q4. "The analysis assumes the upper-level function $F_i$
> is bounded. However, this has not been made in the aforementioned [1,2,3,4] works. More clarifications should be provided. If there are any special challenges making this assumption necessary? Also, the assumption $q(z_k) \le B$ is a strong assumption that has not been made in previous bilevel and MOO literatures, because bounded function value and strong-convexity cannot be made simultaneously."
>
> **A4**: (i) To prove the convergence of gradient norm $||d_k||^2$, we sum the inequality of $||d_k||^2$ over $k = 0, 1,...,K-1$ and obtain the inequality (22) in Appendix. This inequality leads to an upper bounded term $\sum_{k=0}^{K-1}\sum_{i=1}^m(\tilde{\lambda}^k_i -\tilde{\lambda}^{k+1}_i)F(z^{k+1})$. Though the convergence of $\tilde{\lambda}^k_i -\tilde{\lambda}^{k+1}_i$ can be controlled by the momentum parameter $\beta_k$, $F_i(z)$ should be bounded to make the whole term decrease to zero. Therefore, for our method, the boundedness assumption of the upper-level function $F_i$ is necessary.
>
> Since the objective functions are assumed to be convex or strongly convex in [1], the boundedness assumption can be unnecessary in [1]. However, our convergence analysis is based on the non-convex case. The MGDA-based methods [2,3,4] all apply a gradient descent method to update the weights $\lambda$ in each iteration. For solving a standard MOO problem $F(x)$, these methods have technical advantages, as the convergence can be controlled by the learning rate of $\lambda$ and the form of the gradient of $\lambda$ is very simple, i.e., $\nabla F^T \nabla F$ ([2,3] add an additional regularization term, so their corresponding gradient should add a positive constant). Overall, this updated approach of $\lambda$ benefits their convergence analysis and allows them to avoid boundedness assumptions. In contrast, our work focuses on solving an MOBLO problem rather than MOO. The optimization problem of $\lambda$ is problem (10), which is different from that of the MGDA-based MOO methods [2,3,4]. For this optimization problem (10), the expression of the gradient of $\lambda$ is very complex, since we have an additional $\nu\nabla q(z)$ term and there exists a max operator in the objective function. So it is hard to use the strategies in [2,3,4] to remove the boundedness assumption in our method.
>
> Moreover, in the context of non-convex optimization, the boundedness assumption on the objective function value is still commonly employed in both multi-objective [3,5] and single-objective optimization [6] problems. Therefore, we employ this assumption to facilitate the convergence analysis of the proposed FORUM method in addressing the MOBLO problem with a non-convex UL subproblem.
>
> (ii) This boundedness assumption $q(z_k)\le B$ is only made for the generated sequence $z_k$ rather than for the entire $\mathbb{R}^{n+p}$ space in our non-asymptotic analysis. So it can hold with strong convexity of function $f$. When we conduct our algorithm on a local bounded region of $z$ for a Lipschitz continuous function $f$, this assumption can also hold. Therefore it is not a very strong assumption.
>
> ---
>
> **References**
>
> [1] Suyun Liu and Luis Nunes Vicente. The stochastic multi-gradient algorithm for multi-objective optimization and its application to supervised machine learning. Annals of Operations Research, 2021.
>
> [2] Lisha Chen, Heshan Fernando, Yiming Ying, and Tianyi Chen. Three-way trade-off in multi-objective learning: Optimization, generalization and conflict-avoidance. arXiv:2305.20057.
>
> [3] Heshan Devaka Fernando, Han Shen, Miao Liu, Subhajit Chaudhury, Keerthiram Murugesan, and Tianyi Chen. Mitigating gradient bias in multi-objective learning: A provably convergent approach. In International Conference on Learning Representations, 2023.
>
> [4] Peiyao Xiao, Hao Ban, and Kaiyi Ji. Direction-oriented Multi-objective Learning: Simple and Provable Stochastic Algorithms. arXiv:2305.18409.
>
> [5] Shiji Zhou, Wenpeng Zhang, Jiyan Jiang, Wenliang Zhong, Jinjie Gu, and Wenwu Zhu. On the convergence of stochastic multi-objective gradient manipulation and beyond. In Neural Information Processing Systems, 2022.
>
> [6] Kfir Levy, Ali Kavis, and Volkan Cevher. Storm+: Fully adaptive sgd with recursive momentum for nonconvex optimization. In Neural Information Processing Systems, 2021.

---

### Official Review · Reviewer_MMwK · 2023-11-05

**Soundness:** 3 good
**Presentation:** 3 good
**Contribution:** 3 good
**Rating:** 6
**Confidence:** 3

**Summary:**

This paper proposes a novel first-order multi-gradient algorithm called FORUM for solving multi-objective bi-level optimization problems. The proposed method achieves state-of-the-art performance on three multi-task learning benchmark datasets. The paper also provides a reformulation of the MOBLO problem as a constrained multi-objective optimization problem using the value-function-based approach. The proposed method is evaluated through empirical experiments, which demonstrate its effectiveness and efficiency.

**Strengths:**

1. Novelty: The paper proposes a new method, FORUM, for solving multi-objective bi-level optimization problems that is based on a first-order multi-gradient algorithm. This is a novel approach that addresses the computational inefficiency of existing gradient-based methods that require computing the Hessian matrix.

2. Efficiency: The proposed FORUM algorithm is shown to be more efficient than existing methods based on complexity analysis. The paper provides a theoretical analysis of the algorithm's complexity and a non-asymptotic convergence result. Empirical experiments also demonstrate the efficiency of the proposed method in different learning problems.

3. Effectiveness: The proposed FORUM algorithm achieves state-of-the-art performance on three multi-task learning benchmark datasets. The paper provides extensive experimental results that demonstrate the effectiveness of the proposed method in comparison to other state-of-the-art algorithms.

**Weaknesses:**

1. Limited scope: The paper only evaluates the proposed FORUM algorithm on two learning problems, i.e., multi-objective data hyper-cleaning and multi-task learning on three benchmark datasets. The generalizability of the proposed method to other learning problems is not thoroughly explored.

2. Lack of comparison with non-gradient-based methods: The paper only compares the proposed FORUM algorithm with existing gradient-based methods, such as MOML and MoCo. It would be interesting to see how the proposed method compares to non-gradient-based methods, such as evolutionary algorithms or swarm intelligence.

3. Lack of implementation details: The paper does not provide detailed implementation information about the proposed FORUM algorithm, such as the specific hyperparameters used in the experiments. This makes it difficult for other researchers to reproduce the results and compare the proposed method with their own algorithms.

**Questions:**

Regarding the use of approximation methods in the paper, I have a question for the authors. While the paper proposes an approximation method to compute ω∗(α) and approximates the constraint function q(z) using eq(z) = f(z)−f(α, ˜ωT ), it is not clear how the approximation errors affect the performance of the proposed FORUM algorithm. Could you please provide more insights into the impact of the approximation errors on the convergence and efficiency of the proposed method?

---

> ### Author Response · Authors · 2023-11-20
> **Reply to Reviewer MMwK (1/2)**
>
> Thank you for your thoughtful review and constructive comments.
>
> ---
>
> >Q1. "Limited scope: The paper only evaluates the proposed FORUM algorithm on two learning problems, i.e., multi-objective data hyper-cleaning and multi-task learning on three benchmark datasets. The generalizability of the proposed method to other learning problems is not thoroughly explored.''
>
> **A1**: (1) We evaluate the proposed FORUM method on **two** datasets for multi-objective data hyper-cleaning and **three** datasets for multi-task learning, which includes image classification, dense prediction, and molecular property prediction. Extensive experimental results demonstrate FORUM achieves state-of-the-art performance on all datasets.
>
> (2) FORUM is a general algorithm that can be applied to any learning problems taking the optimization form of a multi-objective bi-level optimization problem. We will study more learning problems in the future.
>
> ---
>
> >Q2. "Lack of comparison with non-gradient-based methods: The paper only compares the proposed FORUM algorithm with existing gradient-based methods, such as MOML and MoCo. It would be interesting to see how the proposed method compares to non-gradient-based methods, such as evolutionary algorithms or swarm intelligence.''
>
> **A2**:  Thanks for your suggestion. We compare the proposed FORUM method with two non-gradient-based methods, i.e., the Bayesian method and the NSGA-II method, on the multi-objective data hyper-cleaning problem. The experimental setting is the same as in Section 5.1 of the submission. For these two non-gradient-based methods, in each iteration, we first learn a model $\omega$ by SGD with the given $\alpha$, then compute the upper-level objectives, and finally update $\alpha$ by the non-gradient-based method. We implement the Bayesian method with the open-source Hyperopt library [1]. The NSGA-II method [2] is implemented based on the open-source Optuna library [3]. The experimental results are shown in the following table. The results show that the proposed FORUM is more effective than those non-gradient-based methods.
>
> \begin{array}{lcccc}\hline & \text{\textbf{MNIST} Accuracy} & \text{\textbf{MNIST} F1 Score} & \text{\textbf{FashionMNIST} Accuracy} & \text{\textbf{FashionMNIST} F1 Score} \newline\hline\text{Bayesian method} & 85.70_{\pm0.92} & 85.71_{\pm0.93} & 76.36_{\pm1.93} & 76.17_{\pm2.10} \newline\text{NSGA-II} & 85.79_{\pm1.97} & 85.76_{\pm1.96} & 76.39_{\pm1.95} & 75.78_{\pm2.02} \newline\text{FORUM} (T=16) \text{(\textbf{ours})} & \mathbf{90.79}\_{\pm0.33} & \mathbf{90.79}\_{\pm0.33} & \mathbf{82.37}\_{\pm1.00} & \mathbf{82.10}\_{\pm1.16} \newline\hline\end{array}
>
> ---
>
> **References**
>
> [1] James Bergstra, Daniel Yamins, and David Cox. Making a Science of Model Search: Hyperparameter Optimization in Hundreds of Dimensions for Vision Architectures. In International Conference on Machine Learning, 2013.
>
> [2] Kalyanmoy Deb, Amrit Pratap, Sameer Agarwal, and T. Meyarivan. A Fast and Elitist Multiobjective Genetic Algorithm: NSGA-II. IEEE Transactions on Evolutionary Computation, 2002.
>
> [3] Takuya Akiba, Shotaro Sano, Toshihiko Yanase, Takeru Ohta, and Masanori Koyama. Optuna: A Next-generation Hyperparameter Optimization Framework. In SIGKDD Conference on Knowledge Discovery and Data Mining, 2019.

---

> ### Author Response · Authors · 2023-11-20
> **Reply to Reviewer MMwK (2/2)**
>
> >Q3. "Lack of implementation details: The paper does not provide detailed implementation information about the proposed FORUM algorithm, such as the specific hyperparameters used in the experiments. This makes it difficult for other researchers to reproduce the results and compare the proposed method with their own algorithms.''
>
> **A3**: The implementation details, including specific hyperparameters, are provided in Appendix D: Appendix D.1 for multi-objective data hyper-cleaning (Section 5.1) and Appendix D.2 for multi-task learning (Section 5.2). We will release the code later.
>
> ---
>
> >Q4. "Regarding the use of approximation methods in the paper, I have a question for the authors. While the paper proposes an approximation method to compute $\omega^*(\alpha)$ and approximates the constraint function $q(z)$ using $q(z) = f(z)-f(\alpha, \tilde{\omega}^T )$, it is not clear how the approximation errors affect the performance of the proposed FORUM algorithm. Could you please provide more insights into the impact of the approximation errors on the convergence and efficiency of the proposed method?"
>
> **A4**: (i) We have analyzed the error bound of this approximation in Lemma A.1 in the submission. The approximation error exponential decay w.r.t. the lower-level iterations $T$.
>
> (ii) We have also analyzed the influence of this approximation on the convergence result, as shown in Theorem 4.3 in the submission. The convergence rate of FORUM is of the order $\mathcal{O}(K^{-\frac{1}{4}}+\Gamma(T))$, where $\Gamma(T)$ represents exponential decays w.r.t $T $. Therefore, as $T\to +\infty$, the influence of the approximation error on the convergence rate decreases, and the algorithm converges faster.
>
> (iii) We have conducted experiments to show the effect of the approximation error (i.e., with different $T$), and the results are reported in Table 1 in the submission. The results show that the performance is not so sensitive over a large range of $T$.

---

> > ### Comment · Reviewer_MMwK · 2023-11-23
> >
> > Thanks very much for your detailed clarification.

---

> > > ### Author Response · Authors · 2023-11-23
> > >
> > > Thanks for your reply. We are glad that our reply addressed your concerns.

---

### Author Response · Authors · 2023-11-20
**Reply to all the reviewers**

Dear Reviewers,

Thanks to all the reviewers for their constructive and valuable comments. We have revised the paper accordingly and highlighted the changes in cyan.

We have responded to each reviewer separately. We hope that we have satisfactorily addressed your concerns. Please let us know if there are any further concerns or questions.

Best,

The authors

---

### Meta-Review · Area_Chair_RCkU · 2023-12-05

**Metareview:**

This paper studies multi-objective optimization problem where each objective has a bilevel optimization structure. The proposed approach utilizes two ideas to solve this problem: value-function-based method to solve bilevel problem and momentum update.

There was a lot of discussion, but reviewers' were not convinced with some of the responses. Specifically, the outstanding and critical concern is that the convergence results presented in this paper are not only unconvincing but also weaker compared to existing literature. Therefore, I recommend rejecting the paper and strongly encourage the authors' to revise the paper based on the review comments for re-submission.

**Justification For Why Not Higher Score:**

Significant flaw and weakness in the convergence analysis of the proposed method.

**Justification For Why Not Lower Score:**

N/A

---

### Decision · Program_Chairs · 2024-01-16

Reject